# A variational deep-learning approach to modeling memory T cell dynamics

Christiaan H. van Dorp [ID][1☺], Joshua I. Gray[2☺], Daniel H. Paik[ID][2], Donna L. Farber[2*], Andrew J. Yates[ID][1*]

**1** Department of Pathology and Cell Biology, Columbia University Irving Medical Center, New York, New York, United States of America, **2** Department of Microbiology and Immunology, Columbia University Irving Medical Center, New York, New York, United States of America

☺ These authors contributed equally to this work.
* andrew.yates@columbia.edu (AJY); df2396@cumc.columbia.edu (DLF)

**Data availability statement:** All code and data are available on GitHub (https://github.com/chvandorp/scdynsys.git) and Zenodo (https://doi.org/10.5281/zenodo.15353157).

## Abstract

Mechanistic models of dynamic, interacting cell populations have yielded many insights into the growth and resolution of immune responses. Historically these models have described the behavior of pre-defined cell types based on small numbers of phenotypic markers. The ubiquity of deep phenotyping therefore presents a new challenge; how do we confront tractable and interpretable mathematical models with high-dimensional data? To tackle this problem, we studied the development and persistence of lung tissue-resident memory CD4 and CD8 T cells ($T_{RM}$) in mice infected with influenza virus. We developed an approach in which dynamical model parameters and the population structure are inferred simultaneously. This method uses deep learning and stochastic variational inference and is trained on the single-cell flow-cytometry data directly, rather than on the kinetics of pre-identified clusters. We show that during the resolution phase of the immune response, memory CD4 and CD8 T cells within the lung are phenotypically diverse, with subsets exhibiting highly distinct and time-dependent dynamics. $T_{RM}$ heterogeneity is maintained long-term by ongoing differentiation of relatively persistent Bcl-2$^{hi}$ CD4 and CD8 $T_{RM}$ subsets which resolve into distinct functional populations. Our approach yields new insights into the dynamics of tissue-localized immune memory, and is a novel basis for interpreting time series of high-dimensional data, broadly applicable to diverse biological systems.

## Author summary

After an influenza infection, antibodies provide strong protection—but only against the same strain. Because influenza rapidly mutates its surface proteins, this protection soon becomes ineffective. Memory T cells help control infection by reducing symptoms and transmission. A key advantage is that they often recognize conserved parts of the virus, making their protection more durable across strains. Lung tissue-resident memory

**Funding:** This study was supported by the National Institutes of Health awards U01 AI150680 (AJY and DLF) and R01 AI093870 (AJY). The funders had no role in study design, data collection and analysis, decision to publish, or preparation of the manuscript.

T cells ($T_{RM}$) are especially important, as they remain in lung tissue and respond rapidly to reinfection. However, $T_{RM}$ numbers decline over time and may not last long enough to protect during the next flu season. To better understand this decline, we collected a time series of high-dimensional flow cytometry data from the lungs of influenza-infected mice. We analyzed this data using a combination of traditional clustering, deep learning, and mathematical modeling. Our results show that $T_{RM}$ cells are both phenotypically and dynamically diverse—traits that challenge standard modeling approaches. By integrating machine learning with dynamical systems modeling, we identified distinct $T_{RM}$ subsets and estimated their behaviors over time. This approach offers new insights into the maintenance of lung tissue-resident T cell memory and may inform strategies to improve the persistence of cellular immunity.

## Introduction

The widespread implementation of single-cell high-throughput assays has driven the development of a plethora of computational tools for generating lower-dimensional ('latent') representations of datasets that occupy high-dimensional 'feature' spaces, usually involving the identification of discrete clusters of cellular states or phenotypes [1,2]. Typically, inferences are derived from single snapshots of systems at steady state. However, following the response of a system to perturbations has the potential to reveal more about the components and interactions that underpin it. Mathematical models are used widely for formulating mechanistic descriptions of cellular dynamics, and drawing inferences from time series of observations. Such models usually describe the directly-observed trajectories of small numbers of discrete, pre-defined cell types or states. An outstanding challenge, then, is to connect dynamical modeling tools to exploit the information contained with the burgeoning quantities of high-dimensional data that are now available.

An acute immune response is an archetype of a dynamic, perturbed set of biological processes. In particular, T cells that respond to infectious challenge undergo clonal expansion and diversification, generating heterogeneous populations of memory-phenotype cells that are capable of responding rapidly and potently to re-exposure to the same infectious agent. These memory T cells can be broadly characterized as circulating or tissue-resident [3]. To date, the majority of modeling studies have focused on the dynamics of circulating memory subsets in both humans and mouse models, characterizing their ontogeny, diversity, kinetics, and persistence [4–8]. In contrast, the dynamics of tissue-resident memory T cells ($T_{RM}$), which patrol organs and barrier sites and provide indispensable front-line protection against a wide variety of pathogens [9], are relatively understudied [10]. Understanding how $T_{RM}$ are established and maintained, and what determines the duration of protection that they offer, has the potential to revolutionize vaccine design [11], our understanding of autoimmune responses, and the control of tumor growth.

As with cell types in many areas of biology, $T_{RM}$ are usually defined by combinations of surface markers and are traditionally identified with flow cytometry using hierarchical binary gating strategies [11]. CD8 $T_{RM}$ in the lung are commonly associated with CD103, which binds to E-cadherin present on epithelial cells, supporting retention in barrier tissues, and CD69, which limits tissue egress by antagonizing S1P1-mediated extravasation [9,11]. CD4 $T_{RM}$ in the lung are typically characterized by expression of CD69 and the chemokine receptor CXCR6 [12,13]. However, the expression patterns of many of these markers do not yield objectively defined cutoff values to characterize positive and negative populations unambiguously, a problem compounded as the number of these markers grows [14]. The ontogeny and

persistence of $T_{RM}$ therefore provide an ideal setting to explore the application of dynamical modeling to high-dimensional phenotyping data.

An obvious approach is to perform an initial unsupervised clustering step, which combines dimension reduction and phenotypic classification. One can then use sets of ordinary differential equations (ODEs) or other dynamical systems to describe the time evolution of the sizes of these clusters [15], some of which may represent phenotypes that are not canonically defined. This process typically yields estimates of rates of cell loss, self-renewal, differentiation, and cell killing. In principle this sequential approach is highly tractable, because many off-the-shelf clustering methods are available, and standard tools can be applied to perform inference and model selection. Potential problems remain, however. First, clusters are usually not clearly separable and so in general the assignment of cells to clusters should be probabilistic. Second, as the phenotypic structure of a population that is out of equilibrium evolves, the uncertainty in cluster assignment may also be time dependent. These issues suggest that under some conditions it may be necessary to jointly model the distribution of the flow cytometry data itself, as well as the underlying cellular dynamics.

To explore these issues, we used high dimensional flow cytometry to characterize the dynamics and phenotypic structure of T cell subsets within the lungs of mice that were infected with influenza A virus (IAV) and followed up to 57 days post-infection. To model the time evolution of these populations, we developed and compared two methodologies. The first, 'sequential' approach employs standard clustering methods applied to the processed flow cytometry data pooled across time points, and models the time evolution of the cluster sizes with ODE models. The second, 'integrated' method employs deep learning and stochastic variational inference [16] to simultaneously model the structure and dynamics of observed marker expression, via a lower dimensional representation of the data. With these approaches we find that both CD4 and CD8 T cells within the lung are phenotypically and dynamically heterogeneous. We also identified persistent populations within both subsets that may maintain this diversity over weeks to months through continuous differentiation. We discuss the benefits and limitations of the two approaches to modeling high dimensional time series, and conclude by highlighting areas for further development.

## Results

### The size and phenotypic composition of lung-localized T cell populations change dynamically following influenza infection in mice

The mouse model of influenza infection recapitulates the dynamics of virus morbidity and the immune response seen in human infection, and results in the generation of lung CD4 and CD8 tissue resident memory T cells [17,18] ($T_{RM}$). We followed the T cell responses to influenza A virus (IAV) infection in mice, at timepoints between 6 and 57 days post-infection (DPI; Fig 1A; see Methods). We restricted our analyses to antigen-experienced (CD44$^+$ CD11a$^+$) CD8$^+$ and CD4$^+$ T cells (which hereon we refer to as CD8 and CD4 T cells, for brevity) within the lung tissue, as defined by protection from *in vivo* antibody labeling as described [17,19] (S1 Fig). We previously determined that T cells in this lung-resident niche are highly enriched for influenza-specific memory T cells [19]. Both populations peaked in numbers between 8–10 DPI, shortly after the point of maximum weight loss as a measure of infection morbidity (S2A Fig), and then declined with population half-lives of roughly 3 and 5 days respectively (Fig 1B and 1C). Both CD8 and CD4 T cells approached stable numbers by 37 DPI, after contracting approximately 100-fold and 10-fold, respectively.

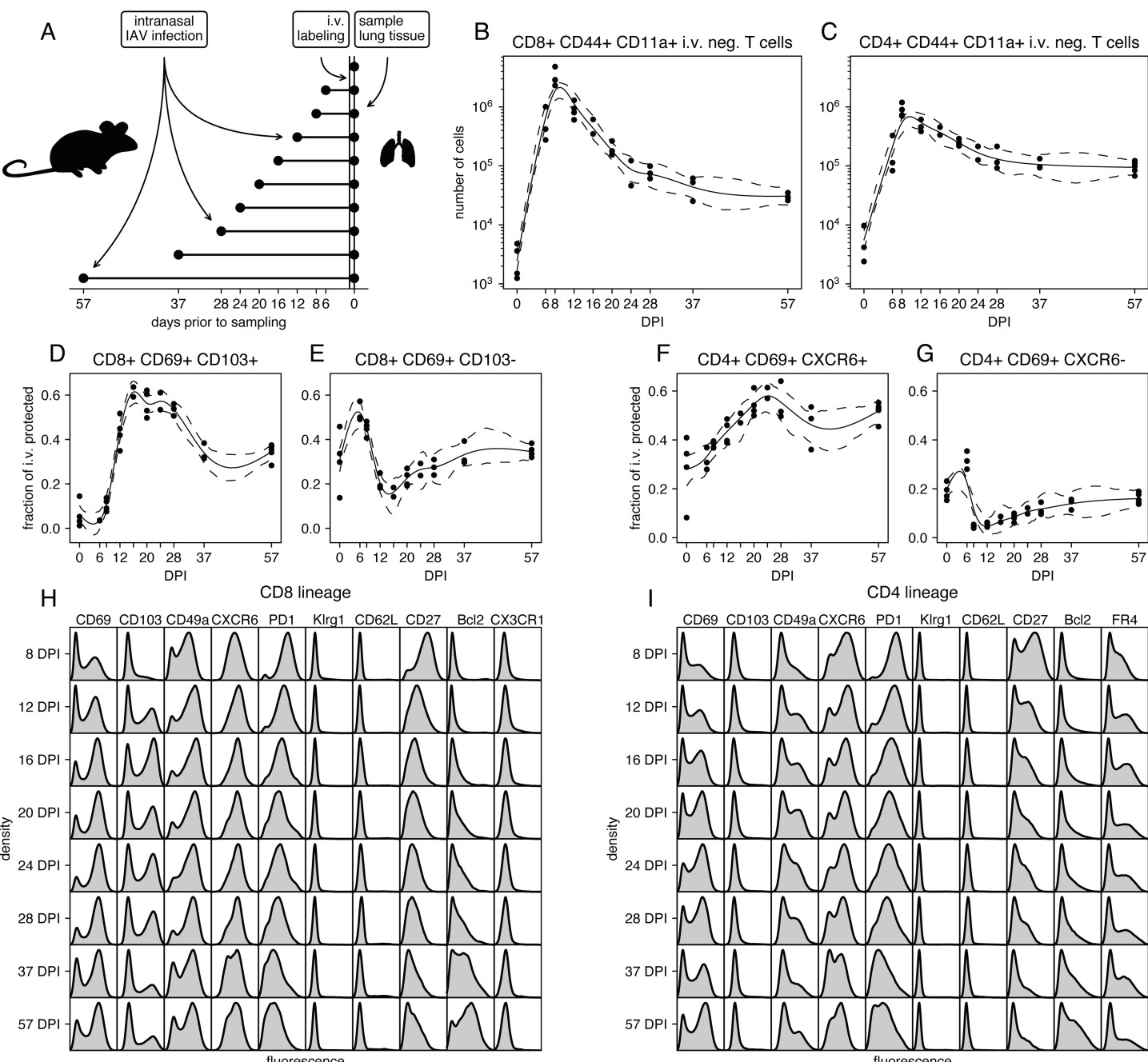

**Fig 1. Studying the timecourse of the lung-localized CD8 and CD4 T cell responses to IAV infection. A.** Experimental design. Each cohort contained 4 or 5 mice and 42 mice were used in total. Eight mice were excluded due to lack of weight loss or i.v. label failure (S2A and S2B Fig). **B–C.** Total numbers of antigen-experienced, protected CD8+ and conventional CD4+ T cells by day post infection (DPI), calculated by multiplying the estimated total number of lung cells with the fraction of protected CD44+ CD11a+ CD8+ and conventional CD4+ T cells. The solid lines (cubic splines) show the average trend, and the dashed lines show 95% confidence intervals (by residual bootstrapping). **D–E.** Frequencies of cells expressing combinations of the tissue-resident memory markers CD69 and CD103 within protected CD44+ CD11a+ CD8+ T cells. **F–G.** Frequencies of cells expressing combinations of the tissue-resident memory markers CD69 and CXCR6 within protected CD44+ CD11a+ CD4+ T cells. **H–I.** Marginal distributions of expression of a selection of markers, aggregated over mice and stratified by day post infection, and lineage (CD8 or CD4).

These smooth trajectories were highly characteristic of adaptive immune responses and of IAV responses in particular [20,21]. However, within them, we identified considerable heterogeneity with respect to surface markers of tissue residency [11,18,20]. The (relative) abundances of T cell subsets defined by combinations of the canonical markers CD69, CD103 and CXCR6 clearly shifted during the contraction phase of the response (Fig 1D–1G). For example, the relative abundance of CD8 T cells expressing both CD69 and CD103 increased with time, then decreased, while CD8 T cells expressing CD69 but not CD103 were rapidly lost in the second week post infection, and then stabilized (Fig 1D–1E). Around half of CD4 T cells expressed both CD69 and CXCR6 beyond 4 weeks post infection. CD4 T cells with low CXCR6 and high CD69 expression mostly disappeared around the peak of the immune response, and then very slowly grew in proportion (Fig 1F–1G). This (temporal) heterogeneity was manifested more clearly when phenotyping was expanded to include markers of cell survival (Bcl-2), access to lymphoid organs (CD62L, typically used to define circulating 'central' memory cells, $T_{CM}$), differentiation (KLRG1 and PD-1), a costimulatory receptor associated with cell survival (CD27), the chemokine receptor CX3CR1, the follicular helper T cell marker FR4, and the integrin CD49a. The marginal distributions of the expression of many of these markers showed clear variation with time (Fig 1H and 1I). Thus, both CD8 and CD4 T cells within the lung exhibit complex and shifting phenotypic structures during the contraction and memory phases of IAV infection.

## Sequential approach

### Standard unbiased clustering methods clearly delineate heterogeneity within CD8 T cells

To understand and quantify how these structures emerge and evolve, we began with a sequential approach to modeling the population dynamics of T cells within the lung from the peak of infection onward. For clarity, we first focus largely on the CD8 T cell response, and then present a more condensed description of the CD4 T cell response that highlights the key results.

We aggregated the flow cytometry data from all mice and all timepoints, and then assigned CD8 T cells to populations using the Leiden clustering method [22] and manual annotation. For each mouse, we tabulated the proportions of cells assigned to each population. Together with the measured total numbers of antigen-experienced protected CD8 T cells in the lung, this process yielded time series of the cluster sizes as a function of time post infection. We used Bayesian dynamical models and the leave-one-out information criterion, implemented in Stan [23,24], to infer biological parameters from these time series and assess the relative support for different models. The workflow of the sequential approach is summarized in Fig 2A.

The Leiden algorithm is stochastic, and we found that both the number and marker expression profiles of the clusters varied from run to run. To address this uncertainty we performed a consensus clustering step. We ran the Leiden method 20 times, and manually annotated the clusters as cell types. Each cell's type was determined by its consensus assignment. We return to the issue of uncertainty in cluster assignment below.

The Leiden clustering robustly identified eight distinct CD8 T cell populations (Fig 2B and 2C). Central memory T cells ($T_{CM}$) were CD62L$^{hi}$ Bcl-2$^{hi}$, and an effector-like population ($T_{Eff}$) was clearly defined by high levels of expression of KLRG1. We identified four $T_{RM}$-like populations based on expression of CD69 and CD103, together with CD49a, CXCR6, and PD-1. We denoted them DP $T_{RM}$ (CD69$^{hi}$ CD103$^{hi}$), CD69 SP $T_{RM}$ (CD69$^{hi}$ CD103$^{lo}$),

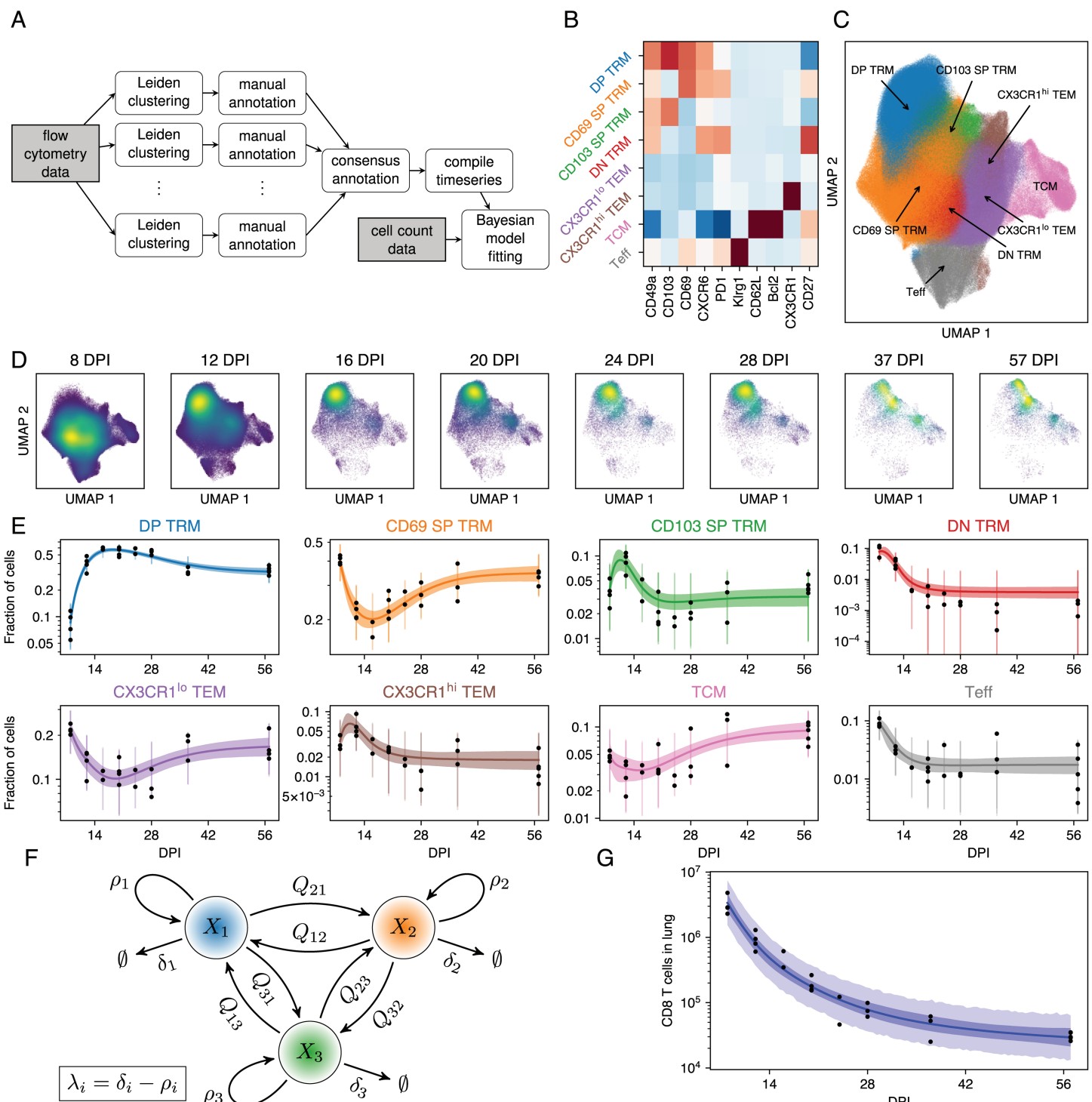

**Fig 2. Pre-processing flow cytometry data for the sequential approach and model fitting.** Results are based on data from *n* = 27 mice. **A.** Flow-chart of the sequential approach. **B.** Marker expression heatmap for selected markers and consensus T-cell populations. **C.** Uniform Manifold Approximation and Projection [25] embedding (UMAP) of the marker expression data, colored by annotation. **D.** UMAPs of marker expression data, split by day post infection (DPI). The color scale reflects cell density in UMAP space. **E.** Time series of the fraction of cells in each cluster (back dots). The lines indicated fitted trajectories from model IV, and the envelopes show the 95% credible intervals (CrI). The vertical bars show the 2.5-97.5 percentiles of the posterior predictive distributions for each data point. **F.** Schematic of the mathematical models, illustrated with 3 populations. $X_i$ is the number of cells in population $i$ and $\lambda_i = \delta_i - \rho_i$ its (possibly time dependent) net loss rate. $Q_{ji}$ is the *per capita* rate of differentiation from $i$ to $j$. **G.** Number of CD8 T cells in the lung (black dots) and the fit (blue line) from model IV. The dark envelope shows the 95% CrI and the light envelope shows the posterior predictive distribution.

CD103 SP $T_{RM}$ (CD69$^{lo}$ CD103$^{hi}$), and DN $T_{RM}$ (CD69$^{lo}$ CD103$^{lo}$), but note that these populations also vary in their expression of other markers. We also identified a population which we define as effector memory ($T_{EM}$) cells, which lacked expression of markers conventionally associated with $T_{CM}$ and $T_{RM}$. A subset of these $T_{EM}$ cells expressed CX3CR1 at high levels, which has been postulated to be a key marker for defining memory CD8 T cell subsets [26] and may indicate a reduced capacity for residency [27].

## IAV nucleoprotein-specific T cells in lung cluster similarly to their polyclonal parent populations

To further validate our clustering step, we used a fluorescently labeled MHC class I tetramer to identify CD8 T cells specific for the NP$_{311-325}$ epitope of IAV, and examined their distribution relative to the cluster locations identified for the response in total (S3C Fig). The tetramer-positive (Tet$^+$) populations represented approximately 10% of CD8 T cells in bulk, and exhibited similar kinetics (S4 Fig), in concordance with previous findings [19]. The partitioning of the UMAP space into 8 subpopulations appeared to respect the density of Tet$^+$ cells (S3C Fig), although they were somewhat enriched within the CD69 SP $T_{RM}$, DN $T_{RM}$ and both $T_{EM}$ populations, suggesting some variation in phenotypic structure with epitope specificity, and/or that a proportion of CD8 T cells within the lung were not IAV-specific. Similar observations held for the NP$_{311-325}$-specific CD4 T cells, identified with an MHC class II tetramer, which represented roughly 5% of total CD4 and were slightly enriched for an effector-like population (S3F Fig). Overall, however, the congruent kinetics of epitope-specific and bulk populations, the fact that these bulk lung niche cells are likely IAV-specific [19], and the improved resolution provided by greater cell numbers, lead us to continue with our analyses of the bulk lung niche CD4 and CD8 T cells.

## Dynamical modeling reveals evidence for ongoing differentiation and time-dependent loss rates of lung CD8 T cell populations

Having partitioned the CD8 T cells among the eight populations, we visualized their time evolution qualitatively by UMAP (Fig 2D) and quantitatively, showing their abundances within the population as a whole (Fig 2E, black dots). The dominant populations were DP $T_{RM}$, CD69 SP $T_{RM}$ and $T_{EM}$, with a relatively stable but lower representation (5-10%) of $T_{CM}$. The $T_{Eff}$ proportion quickly fell during the resolution phase to stabilize at around 1% by 16 DPI, while the DN $T_{RM}$ fraction became almost undetectable over the same timescale. The dynamics of the DP $T_{RM}$ and CD69 SP $T_{RM}$ proportions resembled those derived by manual gating (Fig 1D and 1E), serving as additional validation of the Leiden clustering approach.

To quantify the cellular dynamics underlying these shifts in phenotype, we fitted several models to the timecourses of cluster sizes (Fig 2F). In the simplest model (I), we estimated a single net loss rate (the balance of loss due to cell death and/or egress from the lung, and self-renewal) for each population. In model II we allowed the net loss rates to be time-dependent, reflecting for example changes in inflammation level within the lung after the virus is cleared. In more flexible models we also allowed differentiation between populations, both without (model III) and with (model IV) time-dependent loss. Fits for model IV are shown together with the data in Fig 2E and 2G. To quantify the effect of migration of circulating T cells to the lung during the memory contraction phase, we performed a cell transfer experiment using CD90 congenic mice (S5 Fig and S1 Text Sect A), which suggested that such ingress contributes minimally to the population dynamics after the infection is cleared. In all models we therefore make the simplifying assumption that ingress is negligible.

## The loss of lung-resident CD8 T cells slows progressively following the resolution of IAV infection

From 8 DPI, the biphasic loss of CD8 T cells in the lung (Fig 1C) could in principle be described by model I, in which two or more populations are lost at different but constant rates (S1 Text Sect B). A general feature of this model is that the trajectories of the subpopulation frequencies on a logarithmic scale are necessarily linear or concave, but not convex (S1 Text Sect C). The data were clearly incompatible with this behavior (S6A Fig, first column), and so we rejected it. In model II we allowed the rate of loss of each population to transition between initial and longer-term values at an exponential rate $u$. For simplicity, we assumed that this modulation in dynamics applied to all populations in tandem, potentially reflecting global changes in the lung environment as inflammation is resolved. This model yielded a markedly improved fit (S6A Fig, second column), allowing for convexity in population frequencies (notably CD69 SP $T_{RM}$, $T_{EM}$, and $T_{Eff}$), and better capturing the biphasic decline in total cell numbers. The estimated timescale of transition between the early and long-term dynamics, $1/u$, was approximately 3 days, reflecting the large shifts in relative abundances of phenotypes between 8 and 14 DPI. Formal comparison of models I and II using LOO-IC [28] confirmed strong support for time-dependent loss rates ($\Delta$LOO-IC = $130 \pm 31$; see S1 Text Table A). The long-term loss rates $\lambda_L$ yielded by model II, shown in S6B Fig, are consistent with our prior estimates that influenza-specific CD8 T cells are lost from the lung with a population half life ($\log(2)/\lambda_L$; reflecting the net effect of loss and any self-renewal) of 18-25 days [10].

## Evidence for ongoing differentiation of CD8 T cell subsets within the lung

Model III extended model I to allow differentiation between CD8 phenotypes after 8 DPI. Allowing differentiation between every pair of populations risks overfitting, and so we penalized large values of $Q_{ij}$ by regularizing them with an exponential prior distribution, similar to the lasso [29]. A key prediction of model III is that it allows subset frequencies to stabilize at late time points, as was observed for some populations (S6A Fig), because the balance of differentiation and loss can generate stable equilibria. Indeed we found substantial support for differentiation (model III) over the simplest model with constant loss rates, model I ($\Delta$LOO-IC = $82 \pm 29$). However, time-dependent loss only (model II) was still supported most strongly. We then explored model IV that combined the features of models II and III. This model was able to capture both the rapid early dynamics and long-term stability of subpopulation frequencies (Fig 2E and 2G), and it was very strongly supported statistically ($\Delta$LOO-IC = $29 \pm 6.8$, relative to model II). Indeed, after translating LOO-IC to Bayesian model weights [30], model IV had close to 100% support among the four candidates. However, it predicted differentiation between nearly all subpopulations (S6C Fig), at rates with wide credible intervals. Further, the net loss rates had strong posterior correlations with the differentiation rates, indicating that the model can trade off loss with differentiation. For example, high rates of death and/or onward differentiation, balanced by influx from a precursor, are difficult to discriminate from a more isolated population that is terminally differentiated. This tension between quality of fit and interpretability highlights the limitations of relying on statistical support alone to select models.

## Parameter and model identifiability

The potential trade-off between differentiation and loss described above led us to explore whether the parameters of the model can be identified given the available data. As a first step, we performed a structural identifiability analysis (Methods), which revealed that with perfect

data (i.e. unlimited replicates and dense in time), all model parameters are globally identifiable. Our data are not perfect in this sense, and hence we also explored parameter sensitivity and practical identifiability.

The identifiability of a parameter can be assessed by changing its value by a small amount and observing the changes in the system trajectories (S1 Text Sect D). If none of the trajectories are sensitive to changes in a parameter at any time point, then estimating this parameter reliably may be difficult. We focused on the sensitivity of the trajectories $\pi_k(t)$ and $Y(t)$ to perturbations of the differentiation rates $Q_{ij}$ (S7A, S7B, S8A and S8B Figs). As there are many combinations of $\pi_k$ and $Q_{ij}$, we focused on direct effects: How does changing influx ($Q_{ij}$, black curves) or efflux ($Q_{ji}$, red curves) influence the trajectory $\pi_i$ of T-cell population $i$? As expected, the sensitivity of a trajectory $\pi_k$ where $k \neq i, j$ to a differentiation rate $Q_{ij}$ is generally smaller than the direct effects (S7A and S8A Figs, gray curves). Not surprisingly, increasing the influx from another population increases the relative frequency of the target population, and *vice versa*. However, these effects are most noticeable when the source population is large compared to the target (DP TRM, CD69 SP TRM, CX3CR1$^{lo}$ TEM columns of S7A Fig). This effect can also depend on time. For instance, Bcl-2$^{hi}$ CD4 $T_{RM}$ dominate at late time points, and the trajectories $\pi_i(t)$ of other populations $i$ are highly sensitive to the rate of influx from Bcl-2$^{hi}$ $T_{RM}$ as $t$ approaches 57 DPI (S8A Fig, Bcl-2$^{hi}$ TRM column). Similar effects can be seen when we inspect the sensitivity of the total population size to $Q_{ij}$ (S7B and S8B Figs). From this analysis we conclude that we cannot expect to confidently identify all possible differentiation rates.

Next, we used simulation to investigate the practical identifiability of the differentiation pathways. We fitted model IV (CD8 lineage) and model III (CD4 lineage) to the true data to obtain parameter estimates. We then systematically perturbed the $Q_{ij}$ for each pair of subpopulations ($i,j$): using a range of 51 values between 57% and 175% of the point estimate. With each perturbed parameter value, we then generated a pseudo dataset, and fitted the model to it. We then calculated the correlation between the perturbed, ground-truth parameters $Q_{ij}$ and the values estimated from the pseudo data (S7C and S8C Figs). For the CD8 lineage, we clearly see that the best identifiable rates correspond to efflux from the large DP TRM and CD69 SP TRM populations, which is consistent with our sensitivity analysis. Notice that a small target population size does not preclude identifiability of differentiation rates to that population. For example, we can quite confidently say that the rates of differentiation from DP and CD69 SP TRM to DN TRM are likely small, despite DN TRM being relatively small throughout the time course. The pattern is less clear for CD4 T cells, although we mostly find a positive correlation between ground truth parameters and estimates, and the aforementioned Bcl-2$^{hi}$ TRM population has multiple identifiable efflux rates. Notice that we also find a number of negative correlations between the ground truth and estimates, possibly due to stochasticity.

Examples of how (unperturbed) ground-truth parameters correspond to estimated $Q_{ij}$ values within a simulated dataset are shown in S7D and S8D Figs. Of the $d \times (d-1)$ rates, the ones that are truly large are also estimated to be large, while a clear lack of differentiation is also reflected in a small estimate.

Finally, we investigated the identifiability of the models themselves (I, II, III, IV) as opposed to the parameters. Using the four model fits to the CD8 and CD4 data (S6 Fig; CD4 fits discussed below), we generated pseudo data using the estimated parameters. We then fit all four models to the four pseudo datasets, resulting in 16 fits. For each pseudo dataset, we then used formal model comparison to identify the best model (S7E and S8E Figs). Most noticeably, this analysis shows that if the data is generated by model IV (the most complex), then we would select model IV with very high confidence. However, if III is the true

data-generating model, then we still have a high chance of selecting model IV, especially for the CD8 lineage. Comparing this to S1 Text Table A, we see that for the CD8 lineage, we very confidently identified model IV for the real data, and give virtually no weight to model III. This pattern corresponds to row IV of the model-identifiability matrix (S7E Fig). For the CD4 lineage, we cannot very confidently choose between models II, III, and IV, which corresponds to the row III of the model-comparison matrix (S8E Fig).

In summary, our identifiability and sensitivity analyses show that on the one hand, we cannot expect to confidently estimate all differentiation rates using the available data. On the other hand, when source populations are relatively dominant, their efflux (or lack thereof) into other phenotypes has a large impact on the trajectories of those populations meaning that those differentiation rates can be inferred from our data. Further resolving these issues requires other data sources which we will discuss below.

## Uncertainty in clustering highlights shortcomings of sequential approaches

Stratifying cells within the UMAP by timepoint allows one to visualize changes in the distribution of expression profiles within each cluster (Fig 2D). Ideally, the distribution of cells within a subpopulation should be independent of time, but from 20 DPI onward the distribution within CD69 SP $T_{RM}$ shifted towards $T_{EM}$ and CD103 SP $T_{RM}$, $T_{EM}$ moved towards to $T_{CM}$, and DP $T_{RM}$ moved away from SP and DN $T_{RM}$. Related to these shifts, we found variation in the certainty with which cells could be assigned to each population. We quantified this uncertainty using the empirical entropy of these assignments (S3A Fig). A low entropy corresponds to a cell that is always assigned to the same population, while high entropy indicates that different Leiden runs result in different assignments. Uncertainty of assignment also varied between populations (S3B Fig). These discrepancies indicate that the Leiden clustering method may not capture important features of the data, and suggests that a more refined representation of the high-dimensional data is needed—one that takes into account the time evolution of the cell densities in phenotype space.

## Integrated approach

To deal with uncertainty in cluster assignment, we took inspiration from the single-cell modeling tools scVI [31] and scANVI [32]. We assumed that a cell's flow cytometry phenotype, $x$, can be modeled by a lower-dimensional latent vector $z$ representing the cell's internal state, and that $x$ can be reconstructed from $z$ with a complex non-linear map, modeled with an artificial neural network (NN). The measured fluorescence intensities are noisy, and so $z$ cannot be computed exactly; instead, for a given $x$ we aim to generate a probability density for $z$. To avoid explicitly modeling this density for each cell, we make use of an amortized inference model, using the observations $x$ to predict a parameterized distribution function for $z$. This predictor is also a NN model. In the language of variational autoencoders (VAE), the amortization network is the 'encoder', and the network that generates observations $x$ given a latent state $z$ is the 'decoder' (Fig 3A).

To model the population structure within the latent space, we used a multivariate Gaussian mixture model (GMM) [33,34]. The components and weights of the GMM correspond to the different T-cell subpopulations and their relative sizes, respectively. The ODE models predict the weights at any timepoint, and so the GMM describes a distribution of cell states that changes smoothly with time, parameterized by the rates of loss and differentiation, and the time-independent mixture component parameters. These mixtures overlap within the latent space, and so their time-dependent weights can capture shifting distributions of

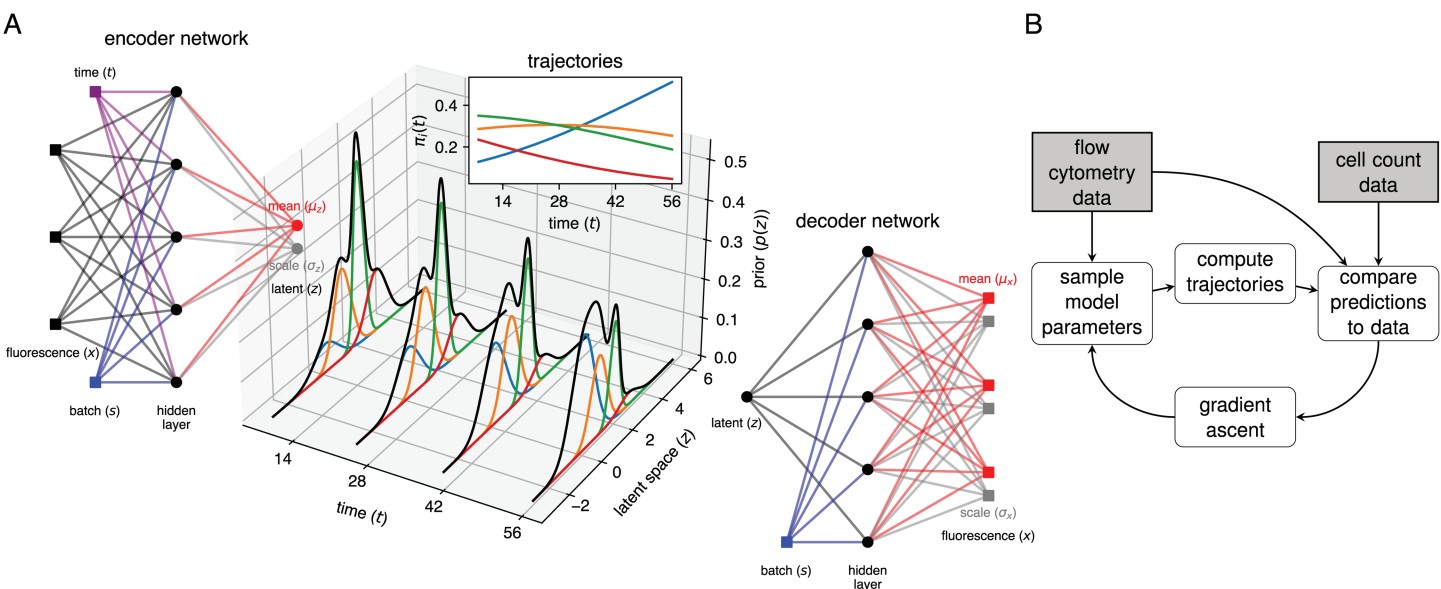

**Fig 3. The integrated approach. A.** Schematic of the VAE model. Using the fluorescence data $x$, we use a fully connected encoder network to infer the latent cell state (specifically, the variational parameters $\mu_z$ and $\sigma_z$ for the posterior distribution of $z|x$; see Methods). The prior distribution of $z$ is given by a Gaussian mixture model with time-dependent mixture weights determined by the dynamical model. The mixture components (multivariate normal likelihood functions) and mixture weights determine the categorical distribution of cluster membership of a cell. Using the decoder network, we compute parameters $\mu_x$ and $\sigma_x$ for the distribution of $x|z$, thereby accounting for the measurement error in $x$. **B.** Flowchart indicating the logic behind the integrated approach. The model is fit directly to single-cell flow cytometry data and cell count data using stochastic variational inference (SVI). The posterior distribution of the model parameters (loss and differentiation rates, initial conditions, NN weights and biases, GMM parameters) are iteratively optimized. In each iteration we sample candidate parameters, and solve the system of ODEs. We then match the data with our predictions, and adjust the posterior to improve this match with the gradient ascent method.

cells in both the latent and feature space. A schematic describing the time-dependent GMM is shown in Fig 3A, where the latent space is represented by the 1-dimensional real line. A model predicting the size trajectories of four subpopulations is used to compute the weights of the mixture components (colored bell curves). The black curves show how the combined latent state distribution of all subpopulations evolves with time. The computational workflow for this method, which we refer to as the integrated approach, is illustrated in Fig 3B. We implemented it in Pyro [35], and describe it in detail in Methods.

## Dimension reduction using the integrated approach preserves information within the flow cytometry data

The sequential approach revealed evidence for both differentiation and time-dependent net loss rates, and so we proceeded with model IV. We first assessed our ability to find a latent representation consistent with our understanding of the marker expression data. We began with a 6-dimensional latent space representation of the 10-dimensional space of marker expression levels. Our results were similar with 5- and 7-dimensional latent spaces. Using a UMAP embedding of the latent space to visualize the learned representation, we saw that the VAE model was able to distinguish cells with different marker expression profiles (Fig 4A) and yielded distinct populations (Fig 4B). To visualize these subpopulations more clearly, we sampled latent vectors from each of the mixture components, mapped them the UMAP space, and calculated the contours of the 75% highest density regions for each of the components. We also took the posterior samples of the location vectors of each of the components and projected them in UMAP space (indicated by colored dots in Fig 4B). These

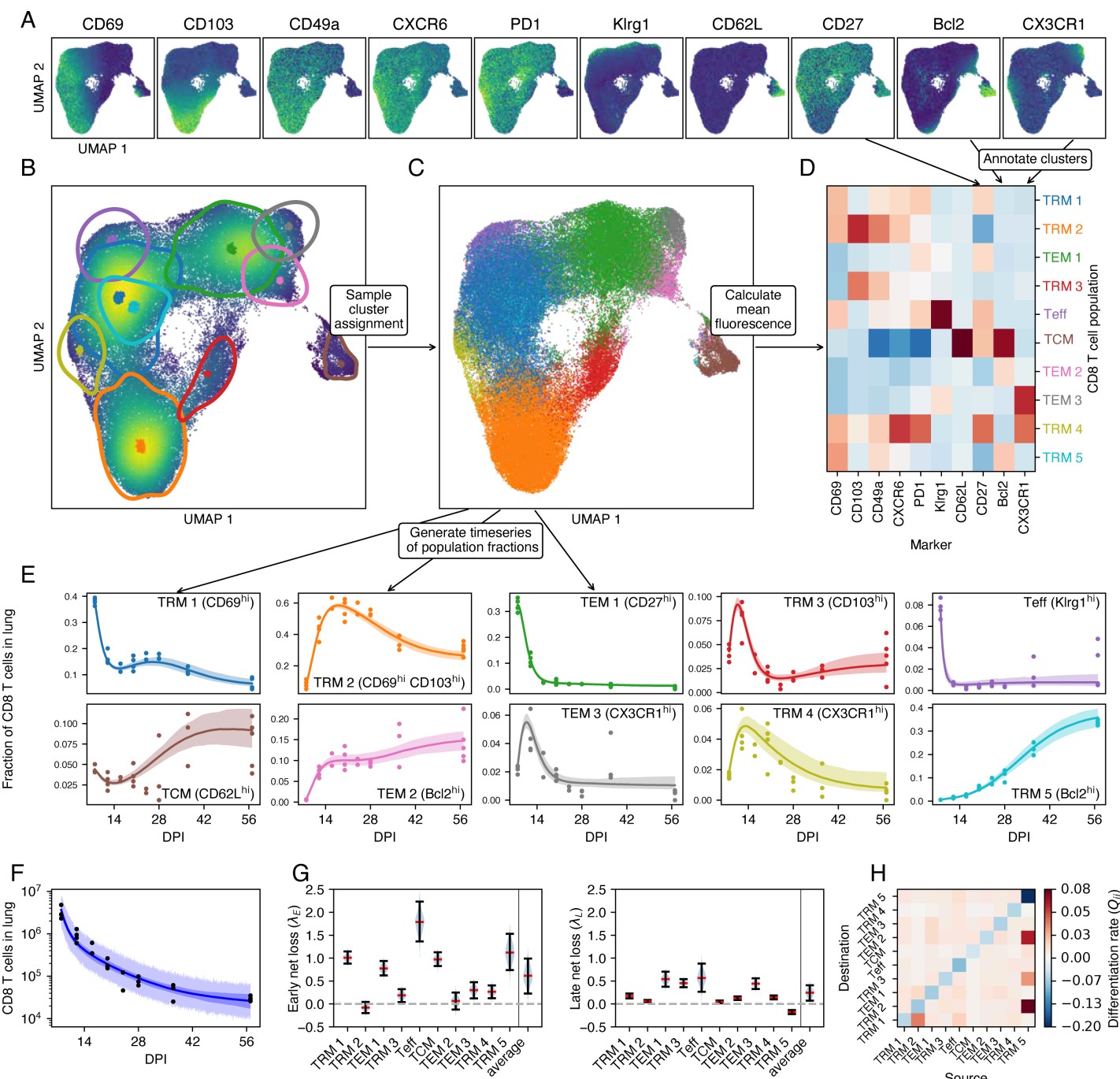

**Fig 4. Modeling the dynamics of lung CD8 T cells with the integrated approach.** Results are based on data from $n$ = 27 mice. **A.** For each marker, the fluorescence intensity of each cell is displayed in a UMAP representation of the latent space. **B.** Cell density and 75% highest density regions for GMM components projected into the same UMAP space. Dots are the GMM location vectors. **C.** Using the GMM and Bayes' formula (Eq 7), a population is stochastically assigned to each cell. **D.** Mean fluorescence intensity of each marker and population identified by the GMM. **E.** Calculated population frequencies (using the assignments drawn in panel C) per mouse and timepoint (dots), with median trajectories predicted by the model (lines) and 95% CrI (bands). For clarity, we indicated some (but not all) highly expressed markers for each population. **F.** Measured numbers of CD44$^+$ CD11a$^+$ protected CD8 T cells in lung (points), trajectory of the median prediction (line), 95% CrI (dark band), 95% posterior predictive interval (light band). **G.** Marginal posterior distributions of net loss rates during the early and late stages, with medians and 95% CrIs. **H.** Differentiation matrix. The diagonal shows the total egress from a population due to differentiation as a negative value.

location vectors can be interpreted as the typical cell state in a component. The contours and typical cell states match well with distinct regions in the latent space, indicating that the GMM is correctly capturing the distribution of latent cell states and, by extension, the distribution of the flow cytometry data. To account for possible batch effects, the batch information was included in the training data for the VAE. However, the trained models did not make use of large batch corrections (S1 Text Sect E), possibly due to the fact that cells from all mice were experimentally analyzed at the same time (Fig 1A).

## The integrated approach yields finer-grained T cell phenotypes than the sequential approach

The GMM defines a distribution of cell states as a function of time, which we can match with data via the decoder NN. However, for each cell we can also use the encoder NN to compute its latent cell state, and then use the GMM to assign the cell to one of the mixture components. This is a probabilistic (or 'soft') cluster assignment; for each cell, the GMM provides a set of probabilities that the cell belongs to each component (given by Eq 7 in Methods). By sampling from categorical distributions constructed in this fashion, we can assign to each cell a single component label (Fig 4C). At this stage, we have effectively clustered the data, but here the key difference with the sequential approach is that this happens *after* we trained our dynamical model. The inference method does not rely at all on the fact that we assign cells to single clusters. Now that we have clustered the flow cytometry data, we can interpret the different mixture components as CD8 T cell subpopulations, and annotate them using the mean fluorescence intensities per cluster (Fig 4D). We compared populations discovered by the two approaches using the Jaccard index (S9A Fig). The integrated approach recovered 7 of 8 populations identified by the sequential method. Namely CD69 SP $T_{RM}$, DP $T_{RM}$, CD103 SP $T_{RM}$, which we denote TRM 1, 2, 3; $T_{Eff}$ and $T_{CM}$; CX3CR1$^{lo}$ $T_{EM}$ and CX3CR1$^{hi}$ $T_{EM}$, which we denote TEM 1 and 3. It did not identify the transient DN $T_{RM}$ subset that expressed PD-1, CXCR6, and the early activation marker CD27, and was lost quickly early in the resolution phase (Fig 2E). Instead, the integrated approach stratified two populations further; CX3CR1$^{lo}$ $T_{EM}$ were separated into Bcl-2$^{lo}$ and Bcl-2$^{hi}$ (TEM 1 and 2), and CD69 SP $T_{RM}$ became three subsets; Bcl-2$^{lo}$ CX3CR1$^{lo}$, Bcl-2$^{lo}$ CX3CR1$^{hi}$, and Bcl-2$^{hi}$ CX3CR1$^{lo}$ (TRM 1, 4, and 5 respectively). This partitioning could not be readily distinguished by UMAP inspection alone (Fig 4A).

## With the integrated approach, data and model are entwined

Having clustered the flow cytometry data, we could again construct time series, enumerate for each mouse the number of cells per population, and then compare their relative sizes (points in Fig 4E) to the trajectories predicted by the fitted model (model IV; curves and 95% credible envelopes in Fig 4E). For convenience, we indicated some highly expressed markers in each panel, but notice that the populations are defined by a complex combination of marker expression. This model accurately captured many features of the distribution of phenotypes, as well as the total population size (Fig 4F). The CD69$^{hi}$ Bcl-2$^{lo}$ $T_{RM}$ (TRM 1) declined rapidly relative to the other populations between 8 and 12 DPI, recovered, but decreased again from around a month after infection, due to the continually increasing prevalence of CD69$^{hi}$ Bcl-2$^{hi}$ $T_{RM}$ (TRM 5). This subset was present initially at a very low frequency, but came to comprise > 30% of lung CD8 T cells by 2 months post infection. A similar pattern can be seen for $T_{EM}$, which increased in frequency soon after the peak of infection but declined rapidly after 10 DPI. The Bcl-2$^{hi}$ $T_{EM}$ (TEM 2) population was initially very small, but quickly replaced Bcl-2$^{lo}$ $T_{EM}$ (TEM 1 and 3) and stabilized at 10% of CD8

T cells from 16 DPI. Therefore the integrated approach reveals that the Bcl-2$^{hi}$ and Bcl-2$^{lo}$ T$_{RM}$ and T$_{EM}$ populations, which are otherwise phenotypically similar, exhibit quite divergent dynamics.

However, there is one subtlety with presenting the data as a timecourse of relative cluster sizes per mouse (the colored dots in Fig 4E). Calculating the probability that a cell is assigned to a cluster uses two pieces of information; the cell's marker expression, and the expected sizes of each of the clusters. The latter are dependent on time, and this time dependence is in turn learned from the flow cytometry data. Therefore, the summary data displayed in Fig 4E are in part generated by the model, and as a result we expect the data points to follow the model trajectories more closely than in a standard fitting approach—in which clustering and dynamic model fitting are executed separately. This apparent diffusion of model and data is only a problem for assessing how well the model fits the data, and does not invalidate our conclusions *per se*. We present more careful posterior predictive checks below.

## Bcl-2-expressing T$_{RM}$ act to sustain lung CD8 T cell populations long-term

As with the sequential approach we found that up to around 14 DPI T$_{Eff}$ are lost most rapidly, followed by TRM 1 and T$_{CM}$. At the same time, TRM 2 and 3, and TEM 2 and 3 were growing in numbers (Fig 4G, left panel). From approximately 14 DPI onward, all populations declined in number except for TRM 5, the Bcl-2$^{hi}$ CD69$^{hi}$ CD103$^{lo}$ subset (Fig 4G, right panel). Bcl-2$^{lo}$ T$_{EM}$ (TEM 1) were lost most rapidly, consistent with their lack of typical residency markers. Notably, at these late times T$_{CM}$ and T$_{Eff}$ were more persistent than canonical T$_{RM}$. Most strikingly, Bcl-2$^{hi}$ CD69 SP T$_{RM}$ (TRM 5) had a positive intrinsic net growth rate in the late memory phase (that is, we inferred that their rate of self-renewal was greater than their combined rates of death or egress from the lung), but their numbers declined due to their high rates of differentiation into DP T$_{RM}$ (TRM 2), Bcl-2$^{hi}$ T$_{EM}$ (TEM 2), T$_{CM}$, and CD103 SP T$_{RM}$ (TRM 3). Therefore, our analyses suggest that Bcl-2$^{hi}$ T$_{RM}$ help to maintain phenotypic diversity among CD8 T cells in the lung in the weeks following IAV infection.

In our sequential analysis, we pointed out identifiability issues with differentiation rates. To address this in the integrated approach, we decided to leverage the latent representation learned by the model. The idea is that distances in the latent space correspond to similarity in cell state, and that state transitions are more likely between similar states. This assumption is commonly made in other approaches to inferring differentiation pathways [36,37]. To implement a similarity-based restriction on differentiation, we used a mixed-effects-type prior distribution for the differentiation rates $Q_{ij}$ that is dependent on the distance between the mixture components of T cell populations $i$ and $j$. The parameter estimate for this mixture model showed that a larger distance is associated with a smaller differentiation rate (S10A and S10B Fig). By inspecting the credible interval of the individual $Q_{ij}$ as a function of the distance between component $i$ and $j$, we found an inverse trend, but also found that the model can account for outliers (S10C Fig). Note that cell surface marker expression measured by flow cytometry is only a limited representation of the cell state.

## Model validation with the integrated approach

In the integrated approach the generated and predicted cluster frequencies are interdependent to an extent. How then do we validate a model's utility or predictive capacity? In contrast to the sequential method, which predicts only the trajectories of the sizes of predefined clusters, the VAE model is generative—that is, it can be used to predict the trajectory of the flow cytometry data itself. Such a trajectory is shown in Fig 5A. Each square shows a UMAP of the latent space at a particular time post infection. The squares highlighted in gray are

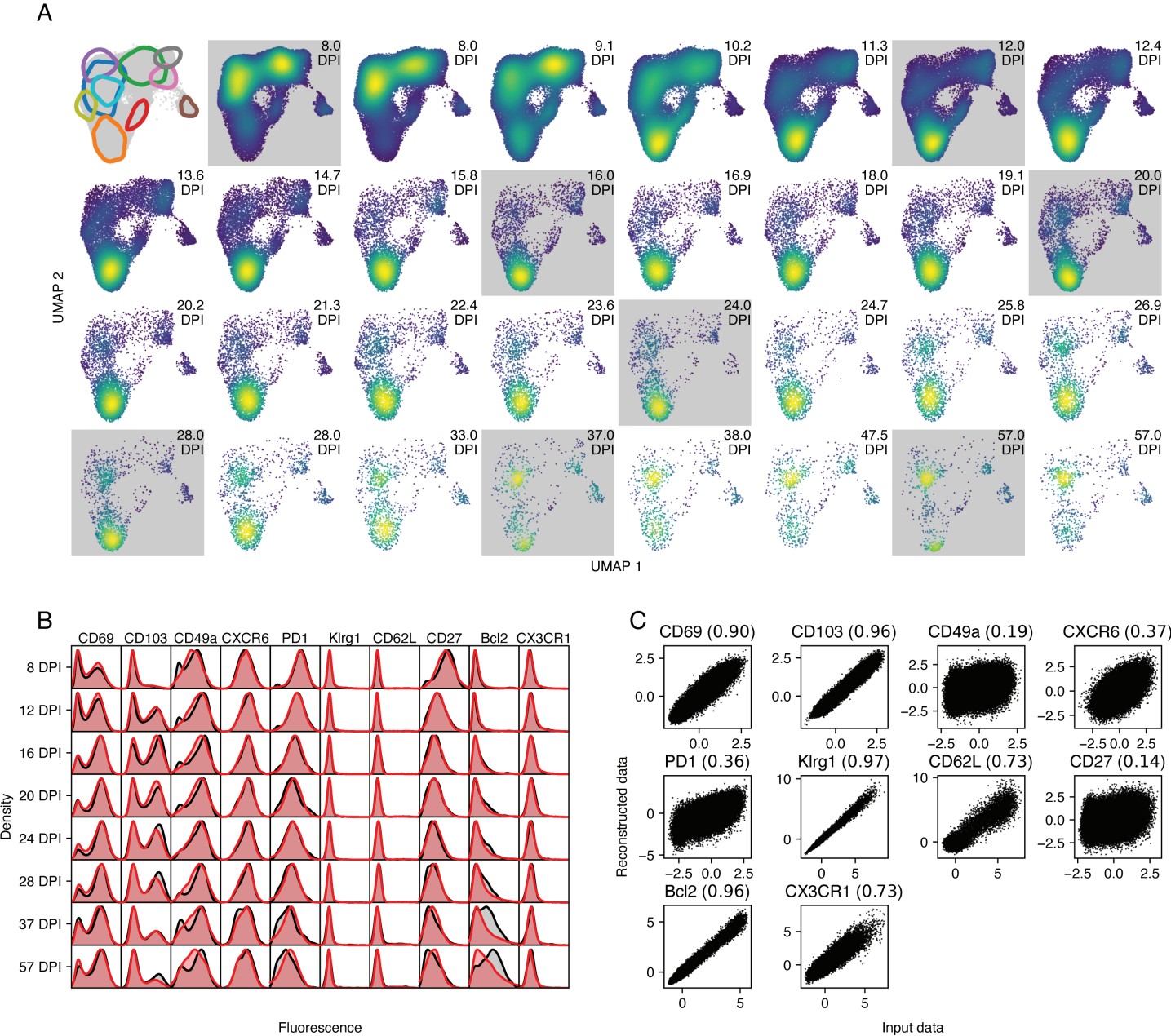

**Fig 5. Posterior predictive checks using simulated marker expression data.** Results are based on data from $n = 27$ mice. **A.** UMAPs of marker expression data, either simulated using the fitted model (white background), or representing the observations (gray background). The location of the inferred clusters are highlighted in the top left panel (colors as in Fig 4B). The color (blue to yellow) shows the density of the points in UMAP space. The number of sampled cells is gradually decreased with DPI to match the sample size of the actual data. The marker expression data (simulated or real) is first mapped to the latent space using the trained VAE model. Then, a UMAP model is trained on the combined true data. The same UMAP model is used to map the simulated data to the plane. **B.** Marginal distributions of marker expression (cf. Fig 1H). Data is shown in black, simulated data is shown in red. **C.** Input data and reconstruction using the autoencoder model. The number in brackets is the coefficient of determination ($R^2$).

UMAPs of the actual data, and the others are simulated by the model. This series of UMAPs acts as a visual posterior predictive check (i.e. a 'model fit'), analogous to a visual assessment of how well the curves in Fig 4B capture the trends in the data points. In this case we can

see the method describes the data well, and highlights time periods that may require denser sampling. For instance, the simulated data between 8 DPI and 12 DPI show rapid shifts in the distribution of phenotypes; CD69 SP $T_{RM}$ (TRM 1) rapidly disappear, DP $T_{RM}$ (TRM 2) increase in frequency, and then $CD27^{hi}$ $T_{EM}$ (TEM 1) start to decline. This order of events is not directly observed, and might be resolved with intermediate samples.

Because the integrated approach models the joint density of the marker expression, we can easily marginalize to single cell surface markers and compare the model predictions with the data. This provides another posterior predictive check (Fig 5B). Pseudo-observations are created by sampling from the GMM at the observation times, and using the decoder network to transform the latent space to the feature space. The marginal distributions of the pseudo observations (red density plots) largely coincide with the true data (black densities), but there are some discrepancies. Most notably, the VAE has difficulties capturing the bimodal shape of the CD49a density, as well as the shape of the Bcl-2 distribution at late time points. To investigate these discrepancies further, we made use of the autoencoder architecture of the model. An autoencoder finds a low-dimensional representation of the data, thereby removing noise and finding essential features [38]. We can test how well our autoencoder has learned to represent the data by first encoding the marker expression dataset, and then reconstructing the expression profiles from the latent cell states using the decoder network. For each marker, we plot the observed fluorescence intensities against the reconstructed values (Fig 5C). A perfect reconstruction would yield a perfect correlation between input and output. For the markers CD69 and CD103, we find a very high correlation ($R^2 = 0.90$ and $0.96$, respectively), while for CD49a and CD27, we find low correlations ($R^2 = 0.19$ and $0.14$, respectively). Hence, it appears that the VAE discarded most of the information encoded by CD49a and CD27 expression, indicating that variation in these markers does not relate strongly to heterogeneity in the dynamics of CD8 T cell subsets in the lung.

## The dynamics of lung-localized CD4 T cells following IAV infection

We then applied the two approaches to the CD4 T cell data. With Leiden clustering and consensus annotation, we identified eleven distinct populations (S11A and S11B Fig). Seven clusters possessed a $T_{RM}$ phenotype, delineated by varying expression levels of canonical $T_{RM}$ markers CD69, CD49a, CXCR6, CD103. One of the $T_{RM}$ subsets also expressed KLRG1, suggesting an effector phenotype. We also identified four populations lacking these residency markers: a $CD62L^{hi}$ TCM-like subset, a subset expressing high levels of FR4 and PD-1, characteristics of T follicular helper ($T_{FH}$) cells in the lung [39], and two populations with low expression of all markers measured, distinguished only by the presence of Bcl-2, which we termed $T_{EM}$.

As we found for the CD8 T cells, the joint distribution of marker expression on CD4 T cells gradually shifted with time post infection (S11C Fig). The peak of the response was dominated by $Bcl-2^{lo}$ $T_{EM}$ and the $CD69^{lo}$ $CD49a^{lo}$ $T_{RM}$ population, which expressed high levels of PD-1 and CD27, markers upregulated during activation (S11D Fig). By 3 weeks post infection, $CD69^{hi}$ $CD49a^{lo}$ and $CD69^{hi}$ $CD49a^{hi}$ $T_{RM}$ dominated, giving way in turn to two Bcl-2-expressing $T_{EM}$ and $T_{RM}$ subsets by 2 months post infection.

We fitted our four model variants to the timecourses of total CD4 T cell numbers (Fig 1C) and the subset frequencies (S11D Fig). Visual inspection and posterior predictive checks indicated that all four models fit the data reasonably well (S12A Fig), but formal model comparisons indicated that time-dependent net loss alone (model II) or differentiation alone (model III) provided sufficient descriptions of the data (S1 Text Table A).

  A variational deep-learning approach to modeling memory T cell dynamics

As was the case for CD8 T cells, all models exposed remarkable heterogeneity in the dynamics of CD4 T cell subsets in the lung after the resolution of IAV infection (S12B Fig; using model II), particularly among the $T_{RM}$-like populations. The loss rates estimated with model III showed a similar, but slightly more homogeneous pattern (S12C Fig). Again using model III, we found evidence for ongoing differentiation from CD69$^{lo}$ CD49a$^{lo}$ $T_{RM}$ to CD69$^{hi}$ CD49a$^{hi}$ $T_{RM}$ and CD69$^{lo}$ CD49a$^{hi}$ $T_{RM}$, and from Bcl-2$^{hi}$ $T_{RM}$ to multiple $T_{EM}$ populations and effector cells (S12D Fig). Finally, the data suggests a pathway from CD69$^{hi}$ CD49a$^{lo}$ $T_{RM}$ to $T_{EM}$. These data indicate the dynamically shifting T cell populations within the local environment, even after resolution of infection. High entropy scores within $T_{RM}$ clusters indicated substantial uncertainty in Leiden cluster assignment for CD4 T cells (S3D–S3E Fig), underscoring the need for integrated analysis to dissect the subtle changes in subset composition.

Next, we used the integrated approach to refine the model fit for the CD4 T cell data. We chose to continue with model III; even though it performed slightly worse than the full model, and was comparable to model II (S1 Text Table A), it enabled us to further explore differentiation pathways without the added complexity and potential lack of identifiability of the larger model. The VAE generated a latent representation that respected marker expression, and the location of the mixture components in UMAP space corresponded well with high density regions (Fig 6A–6B). Using mean fluorescence intensity (Fig 6C), the integrated approach identified $T_{CM}$ (CD62L$^{hi}$) and a KLRG1$^{hi}$ subset that also expressed $T_{RM}$ markers (TRM 7) whose phenotypes were highly concordant with those identified using the sequential approach. In contrast, many of the $T_{RM}$ and $T_{EM}$ populations were distinct between the two approaches (S9B Fig).

The integrated approach identified seven $T_{RM}$ clusters, defined by varying combinations of $T_{RM}$ markers. In contrast to the sequential approach, several of these clusters (TRM 2, 3, 5, 6 and 7) expressed the $T_{FH}$ marker FR4, which was restricted to $T_{FH}$, $T_{EM}$, and $T_{CM}$ in the sequential approach. The remaining $T_{RM}$ clusters (TRM 1 and 4), were presumably canonical $T_h1$-like cells characteristic of a typical IAV response. Again, these subpopulations exhibited diverse loss rates (Fig 6D), with the KLRG1-expressing TRM 7 and $T_{eff}$ lost at the highest rates and TRM 4 (characterized by CD69, CD49a and Bcl-2) showing an intrinsic net positive growth rate.

We found evidence for more flows from FR4$^{lo}$ to FR4$^{hi}$ subsets than *vice versa*, indicating fluxes towards populations with characteristics of T follicular helper cells (TEM 1 to TRM 7 and $T_{FH}$; TRM 1 to TRM 2, TRM 4 to $T_{FH}$ and TRM 6). However, these populations had relatively high loss rates and as the population declined as a whole (Fig 6F) they were outcompeted by Bcl-2-expressing subsets (TRM 4, and TEM 2; Fig 6G), akin to the long-lived $T_{RM}$ in the CD8 data. In contrast to the CD8 lineage, we did not find a relation between distance in the latent space and differentiation rates (S10B and S10D Fig), likely reflecting the tighter clustering of CD4 T cell phenotypes.

As described for CD8 T cells, the generative nature of the VAE provided a visual predictive check (Fig 6H), showing an excellent correspondence between the data and the fitted model. The observed and predicted marginal distributions for surface markers corresponded well (S13A Fig), with the exception of CD49a and CD27. These discrepancies were also noticeable when we reconstructed data using the VAE (S13B Fig) where CD49a and CD27 had the lowest coefficients of determination ($R^2$ = 0.16 and 0.12, respectively). In contrast, CD69, Bcl-2 and FR4 were almost perfectly reconstructed ($R^2 > 0.97$).

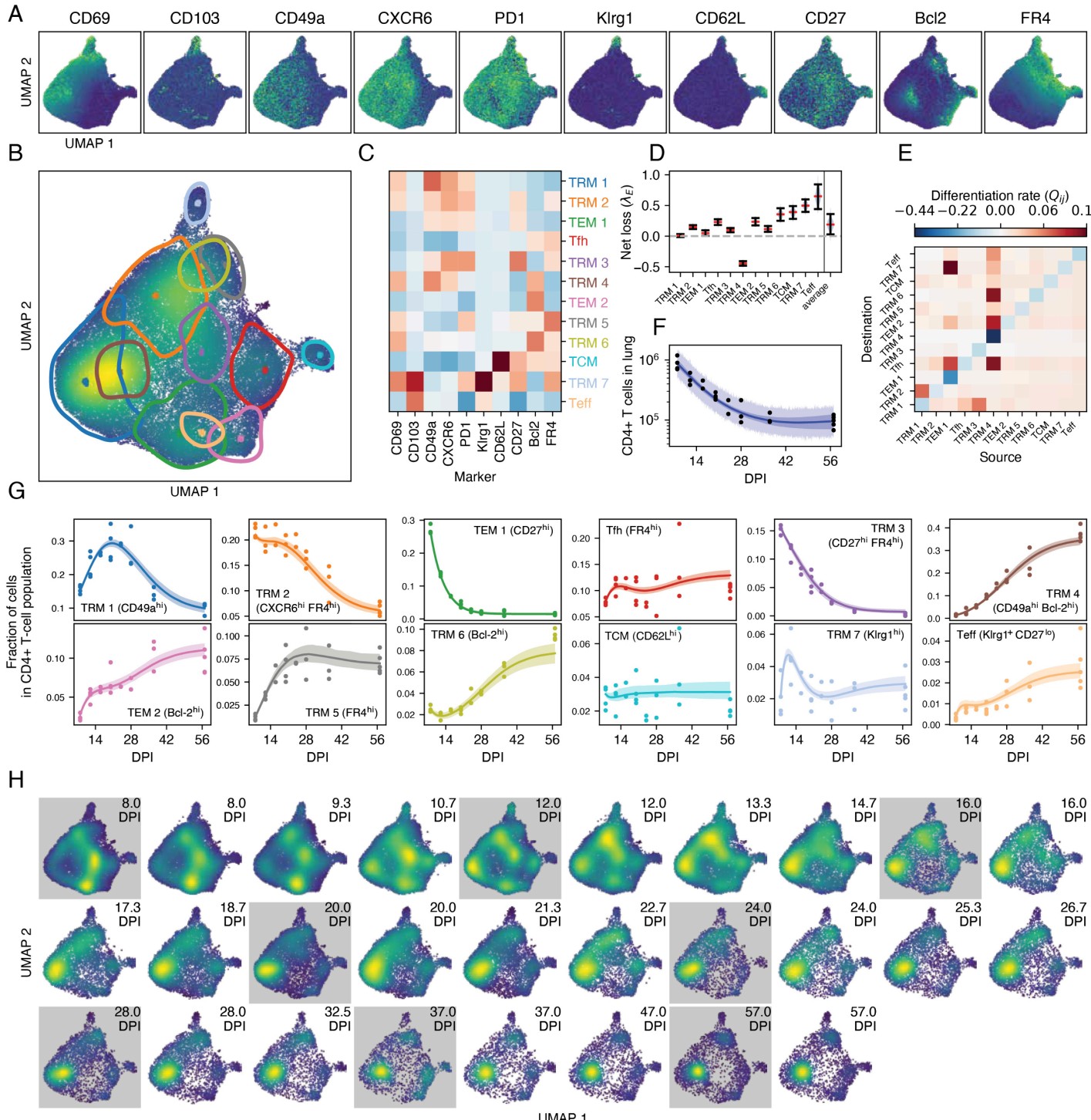

**Fig 6. Modeling the dynamics of lung CD4 cells with the integrated approach.** Results are based on data from $n$ = 27 mice. **A.** For each marker, the fluorescence intensity of each cell is displayed in a UMAP representation of the latent space. **B.** Cell density and 75% highest density regions for GMM components projected into the same UMAP space. Dots are the GMM location vectors. **C.** Mean fluorescence intensity of each marker and population identified by the GMM. **D.** Marginal posterior distributions of net loss rates, with medians and 95% CrIs. **E.** Differentiation matrix. **F.** Measured numbers of CD44$^+$ CD11a$^+$ protected CD4$^+$ T cells in lung (points), trajectory of the median prediction (line), 95% CrI (dark band), 95% posterior predictive interval (light band). **G.** Calculated population frequencies (using assignments drawn using Eq 7) per mouse and timepoint (dots), with median trajectories predicted by the model (lines) and 95% CrI (bands). **H.** UMAPs of marker expression data, either simulated using the fitted model (white background), or representing the observations (gray background).

## Materials and methods

### Experimental procedures

**Mice.** Mice were housed in specific pathogen-free conditions in the animal facilities at Columbia University Irving Medical Center (CUIMC). Eight week old female C57BL/6 were purchased from the Jackson Laboratory. Infections were performed in biosafety level 2 bio-containment animal facilities. All animal studies were approved by Columbia University Institutional Animal Care and Use Committee. Reagents and resources are detailed in SI text Table B.

**Influenza infection.** C57BL/6 mice were briefly anesthetized using inhaled isoflurane and infected intranasally with X31 influenza A virus at a 50% tissue-culture infectious dose (TCID50) of 5000 in 30µl of PBS. Infected mice were weighed daily and animals that lost more than 30% of their starting weight were humanely euthanized. The infections were staggered such that all samples were processed and analyzed on the same day, to mitigate batch effects.

**Tissue preparation.** Mice were intravenously injected with 5µg of Brilliant Violet (BV) 421- conjugated anti-CD45.2 antibody 5 minute before cervical dislocation. Spleen and mediastinal lymph nodes (MedLNs) were processed by mechanical disruption and passed through 70µm filters (Corning). Single-cell suspensions of lungs were prepared first be mechanical disruption followed by digestion with 1mg/ml collagenase and DNAse (Sigma) for 40 minutes at 37°C. Lungs were then passed through a 70µm filter. Red blood cells were lysed from the spleen and lungs using ACK lysis buffer (Gibco).

**Flow cytometry.** Single cell suspensions were washed in PBS before staining with Zombie NIR viability dye (1/1000 in 100µl PBS) for 30 minutes at 4°C. Cells were washed twice and stained with 0.5µl of PE-labeled $IA^b/NP_{311-325}$ (CD4) and 1µl of APC-labeled $H\text{-}2D^b/NP_{366-374}$ (CD8) tetramers (National Institute of Health (NIH) Tetramer Core Facility (NTCF)) in 50µl FACS buffer (PBS with 2% heat inactivated FBS + 0.05mM EDTA). For surface staining, fluorochrome-conjugated antibodies in FACS buffer were added to cell suspensions for 20 minutes at 4°C protected from light, followed by washing with FACS-buffer. Cells were then fixed and permeabilized with the FoxP3 fix/perm kit (Tonbo) for 30 minutes at 4°C. Following fixation, cells were washed with FACS buffer, and stained with antibodies for intracellular detection in Perm Buffer (Tonbo) for 20 minutes at 4°C, followed by washing and resuspension in FACS buffer. Stained cells were acquired using the Cytek Aurora flow cytometer and analyzed using FlowJo version 10 software (Tree Star).

**Congenic transfer experiment.** Three CD90.1 and twelve CD90.2 6-8 week old C57BL/6 male mice were infected with X31 influenza virus (TCID50 5000 in 30 µl). At day 14 of infection, mediastinal lymph nodes and spleen cells were pooled from the three CD90.1 infected mice. From these two groups of pooled cells, T cells were isolated using a Stemcell Technologies EasySep T cell isolation kit. $10^6$ pooled T cells were transferred into the infected CD90.2 mice via intraperitoneal injection and tissue was analyzed at days 2, 4, and 6 post-transfer ($n = 3$ per time point). As a control, $10^6$ pooled T cells from age- and cage-matched uninfected CD90.1 mice were transferred into D14 infected CD90.2 mice and tissue was isolated at 2 days post-transfer ($n = 3$).

### Models

**Sequential approach.** The classical analysis pipeline for our data is as follows. First, the data from all mice are pooled and clustered using a heuristic clustering method. Second, for each mouse, the number of cells in each of the clusters is counted. As the samples were taken

at different times post infection, this results in a time series of cell counts per cluster. Third, dynamical models are fitted to these time series. The resulting parameter estimates and model comparison can give clues about the dynamics of the different T cell populations in the lung.

For the first step (clustering), we used the established Phenograph method [53], which first constructs a $K$ nearest-neighbor graph, and then uses the Leiden algorithm [22] for community detection. The Leiden algorithm itself is a refinement of the Louvain algorithm [54] that finds communities by optimizing a modularity score—the difference of the edge density within and between communities. We ran the algorithm with number of neighbors set to 10 and resolution parameter set to 0.8. We used the implementation from the scanpy package [55]. As the Leiden algorithm is stochastic and depends on the chosen random seed, we repeated the run 20 times and assigned Leiden clusters to more course-grained T cell phenotypes based on marker expression profiles. Each cell was then assigned to the most common phenotype among the 20 iterations. With these assignments, we constructed time series for each phenotype (step 2).

For the third step (fitting models), we used the probabilistic programming language Stan (cmdstan version 2.34) [23,24]. Bayesian inference was performed with the NUTS algorithm, using 5 parallel chains, 500 warm-up iterations and 500 posterior samples. To prevent divergent transitions, we set the "adapt delta" parameter to 0.9.

**ODE models.** We developed a number of candidate models, each making different assumptions about T-cell dynamics and differentiation. The most general model is given in terms of the following system of ODEs

$$\frac{dX}{dt} = -\lambda(t) \circ X + QX \tag{1}$$

with initial condition $X(t_0) = X_0$. Here $X \in \mathbb{R}_+^d$ is a vector of population sizes for each of the $d$ clusters. The vector $\lambda$ contains the net loss rates for each of the clusters, and $\circ$ denotes the Hadamard product (element-wise multiplication). The matrix $Q$ determines T-cell differentiation, and $Q_{ij}$ is the per-capita rate at which cells in cluster $j$ differentiate into cells in cluster $i$. The matrix $Q$ is a stochastic kernel in the sense that the columns add up to zero. Hence, the diagonal elements $Q_{ii} \leq 0$ denote the total efflux from cluster $i$ due to differentiation.

We consider two different forms of the net loss rate $\lambda$. The simplest form is time-independent loss $\lambda(t) = \lambda_E$. This means that the loss rates of each population stays constant over time. However, each population can have its own distinct loss rate. For instance, effector cell counts might decrease faster than $T_{RM}$ or $T_{CM}$ cells.

In addition to a time-homogeneous (or autonomous) model, we also consider a model in which the net loss rate decreases exponentially to a smaller loss rate.

$$\lambda_i(t) = \lambda_{E,i} e^{-u(t-t_0)} + \lambda_{L,i}\left(1 - e^{-u(t-t_0)}\right) \tag{2}$$

The indices $E$ and $L$ refer to 'early' and 'late' stage dynamics. The speed at which this switch happens ($u$) is shared between clusters. In both the autonomous case and the non-autonomous case, we can compare models in which we set $Q = 0$ (i.e. the clusters form independent populations) to models in which we estimate the off-diagonal coefficients of the matrix $Q$ (i.e. allowing for differentiation between clusters).

In the special case when either $Q = 0$ or $\lambda$ does not depend on time, the initial value problem (IVP) given by Eq 1 has a simple closed form solution. This is convenient for inference,

as we will not have to use numerical integrators (and in particular sensitivity or adjoint equations). The solution is given by

$$X(t) = \exp\left(-\int_{t_0}^{t} \text{diag}(\lambda(s))ds + Q(t - t_0)\right)X_0, \quad Q = 0 \vee \frac{d\lambda}{dt} = 0 \tag{3}$$

where exp denotes the matrix exponential, and $\text{diag}(\lambda)$ denotes a square, diagonal matrix with vector $\lambda$ on the main diagonal. The integral of $\lambda$ is given by

$$\int_{t_0}^{t} \lambda_i(s)ds = \begin{cases} \frac{\lambda_{E,i} - \lambda_{L,i}}{u}\left(1 - e^{-u(t-t_0)}\right) + \lambda_{L,i}(t - t_0) & \text{if } u \neq 0 \\ \lambda_{E,i}(t - t_0) & \text{if } u = 0 \text{ and } \lambda_L = 0 \end{cases} \tag{4}$$

In general, we have to resort to numerical integration to solve IVP (1). To integrate ODEs in a manner compatible with the Pytorch and Pyro frameworks, we use the torchode package [56]. Specifically, we use the Dormand–Prince method with the adjoint method for backpropagation. For the sequential approach we use the built-in Dormand-Prince ODE solver in Stan. Although half-lives ($t_{1/2} = \log(2)/\lambda$) are often easier to interpret than loss rates, we chose to present loss rates in figures, as in many cases the 95% CrI contains zero, which would correspond to an infinite half-life.

**Observation (or error) model and prior distributions.** In order to compare our dynamical model to the data, we need to specify a likelihood function. We have two data streams: For each sample (mouse) $s$, we have the number of cells per cluster $i$, denoted $K_{s,i}$. Additionally, we have an estimate of the total number of T cells in the tissue, denoted $M_s$. According to the model, the true cluster frequencies are given by $\pi_i(t_s) = X_i(t_s)/Y(t_s)$, where $Y = \sum_{j=1}^{d} X_j$ denotes the total number of T cells. We then assume that the count vector $K_s$ is sampled from a Dirichlet-multinomial distribution

$$K_s \sim \text{DirMult}(N_s, \pi(t_s)\tau_K) \tag{5}$$

where $N_s = \sum_{i=1}^{d} K_{s,i}$ is the total number of T cells in sample $s$, and $\tau_K$ is a dispersion parameter. Explicitly, the probability of sampling $K_s$ is equal to $N_s B(N_s, \tau_K)/\prod_{i:K_{s,i}>0} K_{s,i} B(K_{s,i}, \tau_K \pi_i(t_s))$, where $B$ is the Beta function. The dispersion parameter $\tau_K$ determines the over-dispersion of the Dirichlet-multinomial distribution, which converges to a multinomial distribution as $\tau_K \to \infty$. We use an over-dispersed distribution because of the significant biological and technical variation between our samples [51].

We assume that the total number of cells in the tissue ($M_s$) has a log-normal distribution

$$M_s \sim \text{LogNorm}(\log(Y(t_s)), \sigma_M) \tag{6}$$

where $Y$ is the total number of cells predicted by the dynamical model. The scale parameter $\sigma_M$ is determined by the technical and biological variation between the experiments.

In addition to a likelihood function, we have to specify prior distributions for the parameters, which are listed in S1 Text Table C. As we do not force the net loss rates $\lambda_E$ and $\lambda_L$ to be positive, the model space includes non-biological instances in which the T-cell population grows exponentially at long time scales. To discourage this, we add a penalty term to the prior distribution. The long-term dynamics is linear and governed by the matrix $A = Q - \text{diag}(\lambda_\infty)$, where $\lambda_\infty = \lim_{t\to\infty} \lambda(t)$. The linear system $dX/dt = AX$ is stable if the spectral

abscissa $\alpha(A) \equiv \max_{i=1,\dots,d}\{\Re(\chi_i)\}$ is negative, where $\chi_i$ are the eigenvalues of $A$. We therefore add a term $-\max\{0, 10\alpha(A)\}^2$ to the log-prior distribution to discourage unbounded growth.

**Model comparison.** For model comparison, we used the Pareto-smoothed importance sampling (PSIS) approximation of leave-one-out (LOO) cross validation [28]. We used the method implemented in the Python package Arviz [57]. The PSIS approximation breaks down for highly influential observations, and Arviz package provides a diagnostic for when this happens. For these observations, we replaced the PSIS approximation with the true LOO-CV statistic by re-fitting the model.

**Parameter identifiability.** We assessed structural identifiability with the Julia package StructuralIdentifiability.jl [58]. To use this package, we first have to cast the system (1) into an autonomous system of ODEs. We observe that $\frac{d\lambda}{dt} = -u(\lambda_E - \lambda_L)\exp(-u(t-t_0)) = -u(\lambda(t)-\lambda_L)$, and hence the extended IVP

$$\frac{dX}{dt} = -\lambda \circ X + QX, \quad X(t_0) = X_0$$
$$\frac{d\lambda}{dt} = -u(\lambda - \lambda_L), \quad \lambda(t_0) = \lambda_E$$

is equivalent to (1). We checked that for $d = 2, 3, \dots, 12$ all parameters (i.e. $\lambda_E, \lambda_L, X_0, u, Q$) are globally identifiable.

Details regarding model sensitivity with respect to the parameters are given in S1 Text Sect D.

**Integrated approach.** We estimate parameters of our ODE model and a set of "nuisance" parameters (NN weights and biases, GMM location vectors) simultaneously in a variational inference framework. We approximate the joint posterior distribution of the latent cell states, the ODE model parameters, and the GMM components with a so-called variational distribution, and optimize the variational parameters using stochastic gradient ascent. This procedure is outlined in Fig 3B. The flow cytometry data is used together with the encoder network to generate a random sample of latent cell states $z$. At the same time, we also sample ODE parameters and GMM location parameters from the variational distribution. Using these parameters, we predict the trajectories of the T-cell population sizes at each sampling time. The total predicted population size is used to compute the likelihood of the total cell counts. The relative population sizes give us the weights of the GMM model, and allow us to compute the prior density for each latent cell state. The decoder network is used compute the likelihood of the flow cytometry data, given the sampled cell states. The prior density and likelihood are combined into the evidence lower bound (ELBO), the objective function of variational inference, which we explicitly define below. Using the gradient of the objective function, the variational parameters are perturbed in the direction of a larger ELBO at which point we can sample new parameters. We repeat this procedure until the ELBO converges.

In our integrated approach, we combine clustering with fitting of the dynamical model in a single machine learning algorithm. We use a variational autoencoder (VAE) to infer a lower dimensional representation $z$ of the marker expression data $x$. We then simultaneously fit a Gaussian mixture model (GMM) to the latent vectors $z$ [33]. Each of the $d$ components of this GMM represents a T-cell cluster. This means that for each data point $x_n$, we can compute the probabilities $p_{n,i}$ that cell $n$ belongs to population $i$ using Bayes' rule

$$p_{n,i} = \frac{\pi_i(t_n)\psi_i(z_n)}{\pi_1(t_n)\psi_1(z_n) + \cdots + \pi_d(t_n)\psi_d(z_n)}. \tag{7}$$

In this equation, $\psi_i$ denote the probability density functions of the Gaussian mixture components, each parameterized with a mean vector $\mu_i$ and covariance matrix $\Sigma_i$, and $\pi_i$ represents the weight of mixture component $i$. In our method, $\pi$ is a function of time and is determined by a dynamical model as in the sequential approach: $\pi(t) = X(t)/Y(t)$. Once we have a fitted model (described below), we can stochastically assign a population to each sample $x_n$ by sampling from the categorical distribution with probability vector $p_n$ (Eq 7). We demonstrate this procedure in Fig 4B and 4C.

The VAE consists of a decoder network $D$ and a encoder network $E$. The decoder network $D : (z, s) \mapsto (\mu_x, \sigma_x)$ takes as an argument a latent vector $z$ and the one-hot-encoded batch information $s$, and returns vectors $\mu_x \in \mathbb{R}^m$ and $\sigma_x \in \mathbb{R}^m_+$ which we can use to sample a vector $x$ in the feature space.

$$x \sim \mathcal{N}(\mu_x, \sigma_x). \tag{8}$$

or compute the likelihood of an observed expression vector $x$, given latent vector $z$. The decoder network $D$ has one hidden layer with 20 nodes and a softplus non-linearity. This hidden layer is shared by the two output layers for $\mu_x$ and $\sigma_x$. We use a softplus link function to ensure that the scale $\sigma_x$ is positive. The encoder network $E : (x, s, t) \mapsto (\mu_z, \sigma_z)$ is similar in architecture to $D$, but takes as an input the expression vector $x$, the batch information $s$, and the sampling time $t$ and returns a location and scale vector for the latent vector $z$, distributed as $z \sim \mathcal{N}(\mu_z, \sigma_z)$. The encoder depends on the sampling time, because it approximates the posterior distribution of $z$, which in turn depends on the time-dependent prior distribution (the GMM).

**Target of optimization: ELBO.** The goal of variational inference is to find an approximate posterior distribution $q$ that is as close as possible to the true posterior, $p$. The Kullback-Leibler (KL) divergence provides a measure of their similarity;

$$D_{KL}(q_\phi(\theta) \| p_\phi(\theta|x)) = \mathbb{E}_{q_\phi(z,\theta)} \left[ \log(q_\phi(\theta)) - \log(p_\phi(\theta|x)) \right]. \tag{9}$$

The posterior distribution is given by

$$p_\phi(\theta|x) = \frac{\pi_\phi(\theta) L(x|\theta, \phi)}{p_\phi(x)}, \tag{10}$$

where $p_\phi(x)$ is the evidence of the model, given the data. We would like to find a parameter vector $\phi$ that maximizes this evidence, but unfortunately $p_\phi(x) = \mathbb{E}_{\pi_\phi(\theta)}[L(x|\theta, \phi)]$ is generally intractable and difficult to approximate directly. This is where the variational distribution is useful. As the KL-divergence is positive, we have

$$\begin{aligned}
\log(p_\phi(x)) &\geq \log(p_\phi(x)) - D_{KL}(q_\phi(\theta) \| p_\phi(\theta|x)) \\
&= \mathbb{E}_{q_\phi(\theta)}[\log(\pi_\phi(\theta) L(x|\theta, \phi)) - \log(q_\phi(\theta))].
\end{aligned} \tag{11}$$

This final quantity is called the evidence lower bound (ELBO). If we then maximize the ELBO, we both minimize the KL-divergence, and given that our variational distribution is close to the posterior, we also maximize the (log) evidence of the model.

To specify the ELBO for our model we have to define three quantities: the prior $\pi_\phi$, the likelihood function $L$, and the variational distribution $q_\phi$. First, the prior factors into three

components

$$\pi_\phi(\theta) = \pi_{\text{GMM}}(\theta_{\text{GMM}}) \times \pi_{\text{Dyn}}(\theta_{\text{Dyn}}|\theta_{\text{GMM}}) \times \pi_{\text{Lat}}(z|\theta_{\text{Dyn}}, \theta_{\text{GMM}}),$$  (12)

corresponding to the GMM parameters, the prior for the dynamical model parameters, and the prior for the latent vectors $z$, respectively. The parameters for the GMM consist of the location (or mean) vectors $\mu_i$ in the latent space of the Gaussian components, and their covariance matrices $\Sigma_i$. The component weights are not free parameters, but are derived from the dynamical model. The location vectors have prior distributions $\mu_i \sim \mathcal{N}_k(0, I_k)$, where $I_k$ is the $k$-dimensional identity matrix. To reduce identifiability issues, all components $i$ have covariance matrix $\Sigma_i = I_k$. Furthermore, let $D_{ij} = \|\mu_i - \mu_j\|_2$ denote the Euclidean distance between location vectors $\mu_i$ and $\mu_j$. To avoid overlap of any two mixture components ($\mu_i \approx \mu_j$), we add the penalty term $-\sum_{i \neq j} D_{ij}^{-2}$ to the log-prior probability.

The priors and parameters $\theta_{\text{Dyn}}$ for the dynamical model are specified in S1 Text Table C. To restrict the space of possible differentiation pathways in an unsupervised manner, we considered a model in which the prior probability of the differentiation rate $Q_{ij}$ is dependent on the similarity between clusters $i$ and $j$. This assumes that cell is most likely to differentiate into a state similar to its original state [36,37]. As a measure of similarity we use the Euclidean distance between the component means in the latent space $D_{ij} = \|\mu_i - \mu_j\|_2$. We take as prior distribution for the differentiation matrix $Q_{ij} \sim \text{Exp}(\mu_Q^{-1} \exp[(D_{ij} - \bar{D})\alpha_Q])$, where $\bar{D}$ is the mean of all distances, and $\alpha_Q > 0$ is an estimated effect of distance on the differentiation rate. Notice that with this parameterization a large distance $D_{ij}$ implies that the differentiation rate $Q_{ij}$ is *a priori* small.

Finally, the prior distribution for the latent vectors $z$ is given by the GMM with density

$$\pi_{\text{Lat}}(z_n|\theta_{\text{Dyn}}, \theta_{\text{GMM}}) = \sum_{i=1}^{d} \pi_i(t_n)(2\pi)^{-k/2}\det(\Sigma_i)^{-1/2}\exp\left(-\tfrac{1}{2}(z_n - \mu_i)\Sigma_i^{-1}(z_n - \mu_i)^T\right).$$  (13)

Here $t_n$ is the time point that cell $n$ was sampled. We assume that the latent vectors of different cells (the $z_n$) are independent. Hence, we ignore any potential dependencies due to common ancestry [59].

The likelihood function $L$ consists of two factors

$$L(x, M|s, \theta, \phi) = L_{\text{Expr}}(x|s, \theta, \phi) \times L_{\text{Count}}(M|\theta, \phi)$$  (14)

which are the likelihood of marker expression data $x_n$, and the likelihood of cell counts $M_n$, respectively. For the likelihood of the expression data, we start with a latent vector $z_n$, and pass this through the decoder network $D_\phi$ to get parameters for the likelihood of $x_n$:

$$x_n \sim \mathcal{N}_m(\mu_x, \sigma_x), \quad (\mu_x, \sigma_x) = D_\phi(z_n, s_n).$$  (15)

The likelihood of the count data is given by

$$M_n \sim \text{LogNorm}(Y(t_n), \sigma_M).$$  (16)

Finally, in order to perform inference, we have to specify a variational distribution $q_\phi(\theta)$. We assume that the variational distribution can be factorized as

$$q_\phi(\theta) = q_{\mathrm{Dyn}}(\theta_{\mathrm{Dyn}}|\phi_{\mathrm{Dyn}}) \times q_{\mathrm{GMM}}(\theta_{\mathrm{GMM}}|\phi_{\mathrm{GMM}}) \times q_{\mathrm{Lat}}(z|\phi_{\mathrm{Lat}}, x), \tag{17}$$

with factors for the dynamical model parameters, the GMM parameters and the latent vectors, respectively. For the (possibly transformed) dynamical model parameter vector

$$\theta_{\mathrm{IVP}} = (\lambda_E, \lambda_L, \log(u), \log(Q), \log(X_0), \mu_{\lambda_E}, \log(\sigma_{\lambda_E}), \mu_{\lambda_L}, \log(\sigma_{\lambda_L}), \mu_Q, \alpha_Q), \tag{18}$$

and $\sigma_M$, we assume a multivariate normal and log-normal distribution, respectively:

$$\theta_{\mathrm{IVP}} \sim \mathcal{N}(\mu_{\mathrm{IVP}}, \Sigma_{\mathrm{IVP}}), \quad \sigma_M \sim \mathrm{LogNorm}(\mu_{\sigma_M}, \sigma_{\sigma_M}). \tag{19}$$

Next, we use a multivariate normal distribution for the location vectors of the mixture components;

$$(\mu_1, \dots, \mu_d) \sim \mathcal{N}_{kd}(\mu_{\mathrm{GMM}}, \Sigma_{\mathrm{GMM}}). \tag{20}$$

Finally, we have to specify a variational distribution for $z$. As we have a latent variable $z_n$ for every observed marker expression vector $x_n$, we use the method of amortization; instead of specifying variational parameter for each individual variable $z_n$, we define a function $E$ that calculates variational parameters $\sigma_z$ and $\mu_z$, based on the data point $x_n$:

$$z_n \sim \mathcal{N}_k(\mu_z, \sigma_z), \quad (\mu_z, \sigma_z) = E_{\phi_{\mathrm{Lat}}}(x_n, s_n, t_n). \tag{21}$$

The function $E$ is defined in terms of a neural network.

We implemented our method in the Pyro language for deep probabilistic programming [35]. We fitted the model to our data using stochastic variational inference (SVI). We used the Adam optimizer [60] with learning rate $2 \times 10^{-3}$ and $10^5$ epochs, and used 20 parallel particles to evaluate the $\nabla$ELBO. The neural network weights and biases were $L_2$-regularized. All our code is available on https://github.com/chvandorp/scdynsys.git.

## Discussion

Developing methods to connect time series of high dimensional data with mechanistic, dynamic models of cell behavior is a pressing challenge. Here we described a straightforward 'sequential' approach in which one clusters and then models the occupancy of these clusters over time. Intuitively, though, this approach may be problematic when phenotypic diversity within a population does not permit clear separation of discrete subpopulations. To address this issue we explored a more holistic approach which allows for time-varying uncertainty in the assignment of cells to clusters, simultaneously modeling the cellular dynamics and its low dimensional representation.

In the setting of the T cell response to influenza A virus infection, the sequential approach showed that the net loss rates of CD8 T cells within the lung are strongly dependent on their phenotype and on time, and also found evidence for ongoing differentiation between cell types. The integrated approach partitioned CD8 T cell phenotypes and their dynamics in a similar way, but also provided more resolution. Specifically, we found that lung-resident CD8 T cells appear to be maintained by a Bcl-2$^{\mathrm{hi}}$ CD69$^{\mathrm{hi}}$ subset that lacks expression of the canonical T$_{\mathrm{RM}}$ marker CD103. Expression of Bcl-2 may be important for counteracting

pro-apoptotic signals arising from the nutrient-poor environment of the airway [40]. It has been suggested that CD69$^{hi}$ CD103$^{hi}$ T$_{RM}$ in the airway are maintained by T$_{RM}$ in the lung parenchyma and that migration is mediated by increased CXCR6 expression [12]. Although here we did not perform bronchoalveolar lavage (BAL) to study cells within the airway directly, we did find evidence for differentiation within the parenchyma from these Bcl-2$^{hi}$ CD69$^{hi}$ T$_{RM}$ into CD69$^{hi}$ CD103$^{hi}$ (DP) T$_{RM}$, which also co-expressed CXCR6 at intermediate levels.

We found that CD4 T cells within the lung were also dynamically and phenotypically heterogeneous, but with subsets less well defined than their CD8 T cell counterparts. In contrast with the CD8 results, the CD4 T cell data could be explained by either time-dependent loss of independent populations, or constant loss of differentiating populations. As with CD8 T cells, however, we found that Bcl-2-expressing populations come to dominate the CD4 T cells within the lung at later time points, and that these cells might be a source for other populations, thereby sustaining heterogeneity. Notably, we also saw evidence for one-way differentiation from canonical T$_{RM}$, which were largely T$_h$1-like, to FR4-expressing cells that were more T$_{FH}$-like.

Our flow cytometry panel contained five markers commonly associated with tissue residency (CD69, CD103, CD49a, CXCR6, and PD1), and others that are typically not expressed on canonical T$_{RM}$ (CD62L, CX3CR1, and KLRG1). T$_{RM}$ have been sub-classified further with markers such as CD101, CXCR3, and CXCR4 [14], and several others [41]. Expanding the panel will likely reveal more fine-grained subpopulations, each with distinct dynamics. For instance, it has been shown that T$_{RM}$ in the airway rapidly lose their CD11a expression [12]. Airway T$_{RM}$ also show distinct dynamics as they are lost more rapidly than T$_{RM}$ in the parenchyma [40]. Here, we used CD11a to pre-select antigen-experienced cells in our gating strategy. If data from BAL were included, adding CD11a to the training data instead could be beneficial. Our inferences could also benefit from inclusion of the Ki67 marker, which is expressed transiently after cell division. Using this marker could allow us to disentangle net loss rates ($\lambda$) into division ($\rho$; Fig 2F) and death or egress ($\delta$). However, this would require more complex models, and will be left for future work.

A typical workflow may benefit from using both the sequential and integrated methods. The sequential approach makes it easy to visualize the data as a set of time series, and hence aids in the the model development phase, and may be sufficient if the subsets are very clearly defined (such as the T$_{CM}$ populations we identified here). The integrated approach is particularly powerful when standard clustering approaches do not clearly delineate phenotypes. In any case it is well suited for fine tuning of the model, parameter estimates and the population structure. Here we used the sequential approach for model selection and explored the integrated approach only with the favored models for CD8 and CD4 T cell dynamics. As the integrated approach is likelihood-based it should also lend itself to model comparison, but performing this is difficult for variational methods and remains an open challenge [42]. Facilitating model selection within the integrated approach would provide significant advantages. In the sequential approach, it is not possible to apply standard selection criteria to establish the optimal number of clusters, because the clustering step occurs before model fitting, and so different numbers of clusters represent entirely distinct datasets. In contrast, the integrated approach could provide an objective measure of the number of subpopulations present in the data, because it deals with the likelihood of the observations themselves.

Increasing the number of clusters in the integrated approach resolved new and distinct populations, such as a CX3CR1$^{hi}$ effector-like subset, and a Bcl-2$^{hi}$ subset of the CD69$^{hi}$ CD103$^{hi}$ T$_{RM}$ within the CD8 lineage; However this increase also generated very small and likely spurious populations that were present at single timepoints, possibly due to batch

effects. Rational determination of the optimal number of clusters, while avoiding spurious populations in an unsupervised manner, is therefore an important future research direction.

A related issue with the integrated method is that, because models are fitted to high-dimensional data, it is challenging to perform posterior predictive checks. Comparing predicted with observed time series can be misleading, as the seemingly observed time series are in part generated by the model. We expect this to be a common issue when dynamic models are fit to more complex data streams. We used two-dimensional UMAP projections to address this issue (Fig 5A), comparing the density of the observations in UMAP space with the density of pseudo-observations generated by the model at intermediate time points. A recent analysis of dimension reduction methods argued that for high-dimensional single-cell data the density in UMAP space corresponds poorly with cell density in the feature space [43]. However, the dimensionality of our data is an order of magnitude lower than the data used in their analyses. Therefore we are confident that, in the realm we explored here, using UMAP for posterior predictive checks is a reasonable procedure.

In both approaches, the model equations take the same functional form for all populations. In general this constraint will be too stringent. For instance, during the expansion phase of the immune response diversifying subsets of T cells may exhibit highly distinct dynamics, which may need to be modeled with structurally different equations. Further, programmed bursts of division characteristic of clonal expansion phase require multiple equations to describe a single population, one for each round of division [44,45]. Structural heterogeneity within models is easily accommodated by sequential approaches, because one assigns phenotypes to each of the clusters before model fitting. It is more problematic for the integrated approach; because we do not know *a priori* which cells will end up in which mixture component, an equation may end up describing the wrong cluster of cells. A potential solution is to use semi-supervised clustering [32]. Here we assign to a small set of cells a phenotype that we want to model with a particular type of equation. We then add a term to the likelihood function that makes it much more likely that those selected cells end up in the appropriate cluster.

Our approach is similar to scVI [31] and scANVI [32] in its use of a VAE, although our method is currently designed for flow cytometry data instead of very high dimensional single-cell RNA sequence data, which requires more complex error models. The use of hierarchical Gaussian mixture models for the analysis of flow cytometry data has been explored elsewhere [46], but here we fit a GMM on a lower-dimensional state space after applying a nonlinear transformation. The transformed data might be more easily described by a GMM than the raw flow cytometry data, especially when the latter is high-dimensional. Autoencoders for single-cell data analysis were first introduced as a de-noising and imputation tool [47], but have evolved rapidly. Currently there exists a large collection of methods based on variational autoencoders [31] which can analyze scRNA-seq data and other modalities. These methods can roughly be grouped into 'exploratory' and 'guided' [48]. Exploratory methods do not use any covariates of interest to guide latent state recovery or community detection, while guided methods explicitly incorporate such data [49]. Our integrated method falls into the guided class of models, as we use time information directly in the data manifold reconstruction, as well as community detection steps.

For the combination of models and data we studied here, parameter identifiability was a legitimate concern. We are trying to infer both the differentiation rates between populations and their net loss rates, from time series of their sizes. Although we showed that with sufficient data these quantities can be distinguished, in practice not all are equally identifiable. In particular, it can be difficult to reliably quantify the flows from small T cell

populations into larger ones. Although this limitation is true for both the sequential and integrated approach, for the latter we have demonstrated a method for leveraging the structure of the latent space to limit the combinatorial complexity of possible differentiation pathways. This method appeared to work for the CD8 lineage, but not the CD4 lineage. For these reasons, the $T_{RM}$ differentiation pathways identified by this study should be verified with further experiments: One could put constraints on possible pathways with additional data from fate-mapping mouse models [50,51] or with developmental trajectories inferred from CITE-seq data [37,52]. Including such data in the presented inference workflows is again an interesting research direction.

Understanding the dynamical properties of immune responses is key to solving many fundamental problems, ranging from finding cures to chronic diseases such as HIV-1 infection, designing vaccines that provide heterosubtypic immunity to respiratory viruses such as influenza, and immuno-therapies targeting tumors. Continuing technological advances are yielding bountiful data at the single-cell level, resolving phenotypes at unprecedented scales and promising to advance our abilities to tackle these problems. Meanwhile, the deep-learning revolution is presenting us with an expanding suite of tools to process, visualize, and interpret these data. Mathematical modeling has a long tradition of aiding the interpretation of immunological and virological data, but has largely relied on heavily curated datasets. The approaches we present in this paper, and others like them, bring dynamical models closer to the observations. This bridging will increase the utility of mathematical models for shaping and testing our intuitions regarding biological processes.

## Supporting information

**S1 Text. All supporting text and tables.**
(PDF)

**S1 Fig. Gating Strategy for CD8$^+$ and conventional CD4$^+$ antigen experienced T cells residing in the lung.**
(PDF)

**S2 Fig. Weight curves of mice after IAV challenge and assessment of I.V. labeling.** Each cohort contained 4 or 5 mice. In total data from 38 mice is presented. **A.** The curves indicate the percentage of the weight relative to that on the day of infection. The red curves correspond to mice that were excluded from further analysis due to lack of weight loss. **B.** Distribution of the I.V. label (CD45.2 IV) for each mouse. The red curves correspond to mice that were excluded from further analysis due to poor I.V. labeling.
(PDF)

**S3 Fig. Validation of population assignment in the sequential approach.** Results are based on data from $n = 27$ mice. **A.** Entropy of CD8 T cell population assignment based on 20 Leiden clustering runs with different random seeds. Black dots corresponds to very certain assignments, yellow dots to highly uncertain assignments. The colored contours indicate the location of the different subpopulations in the UMAP. **B.** Entropy distribution per cluster. The bar plots show the median, IQR and 2.5–97.5 percentile range. The color of the bars and labels correspond to the contours in panel A. **C.** Distribution of IAV NP-specific CD8 T cells in the UMAP with contours indicating clusters. Gray dots indicate bulk antigen-experienced cells. **D–F.** Same as panels A–C, but for CD4 T cell data.
(PDF)

**S4 Fig. Timecourses of NP-specific T cells.** Results are based on data from $n = 34$ mice. **A.** Number of NP-specific CD8 T cells, as a function of post-infection sampling time. The curve represents a spline fit to the log-transformed T cell counts, and the dashed lines represent the 95% confidence envelope (estimated by bootstrapping residuals). The red curve indicates the number of polyclonal CD8 T cells (cf. Fig 1B). **B.** The Tet$^+$ fraction of bulk CD8 T cells in the lung niche. Splines are fitted on the logit scale. **C.** Number of NP-specific CD4 T cells. **D.** The Tet$^+$ fraction of bulk CD4 T cells.
(PDF)

**S5 Fig. Congenic transfer experiment to assess the amount of ingress during the memory phase.** Results are based on data from $n = 12$ mice in total, with 3 mice per group. **A.** Design of the congenic transfer experiment. **B.** Numbers of protected and labeled, antigen-experienced CD8 and CD4 T cells from host and donors combined (upper panels) and donors only (lower panels). We fitted a log-linear model to the cell counts (solid line: ML estimate, dashed lines: 95% confidence envelope). The indicated $\lambda$ is the estimated net loss rate ($\pm$ standard error).
(PDF)

**S6 Fig. Modeling the dynamics of lung CD8 T cells with the integrated approach.** Results are based on data from $n = 27$ mice. **A.** Top panels: numbers of CD8$^+$ T cells in the lung (black dots) and the model fits (blue lines). Remaining panels: observed and predicted sub-population frequencies. Bands indicate 95% credible envelopes, bars indicate the 2.5 and 97.5 percentiles of the posterior predictive distributions. **B.** For model II, marginal posterior distributions of the parameters as violin plots (blue), with median (red) and 95% credible intervals (CrI; black). **C.** Differentiation rates in model IV. Diagonal; total egress rates by differentiation are represented by negative values.
(PDF)

**S7 Fig. Identifiability and sensitivity analysis for the CD8 T cell lineage. A.** Sensitivity of relative cluster sizes ($\pi_k(t)$) with respect to differentiation rates $Q_{ij}$. Each plot contains $d = 8$ curves $\partial \log(\pi_k(t))/\partial Q_{ij}$. The primary effects are highlighted in black ($i = k$) and red ($j = k$), while the secondary effects are shown in gray. The $y$-axes are shown on a square-root-scale. **B.** Sensitivity of total population size ($Y(t)$) with respect to differentiation rates $Q_{ij}$. The curves correspond to $\partial \log(Y(t))/\partial Q_{ij}$. **C.** Practical identifiability scores of differentiation rates $Q_{ij}$. The heatmap shows the correlation coefficient between the ground truth value of $Q_{ij}$, and the estimated value. The stars indicate the levels of statistical significance ($\cdot \, p < 0.1$, $^* p < 0.05$, $^{**} p < 0.01$, $^{***} p < 0.001$) based on a range of 51 ground truth $Q_{ij}$ values. **D.** A single pseudo-dataset is simulated with model IV, using parameters estimated from the true data. Model IV is then fit to the pseudo-data, and for each pair of populations ($i,j$) we show the estimate versus the ground truth value of $Q_{ij}$. **E.** Model identifiability. Data was simulated with and fit to each of the four models, resulting in 16 model fits. For each simulated dataset, the four model fits are compared with model weights (shown in color). High diagonal model weights indicate that the ground-truth model is correctly identified. Shown is the median of 3 simulations for each model.
(PDF)

**S8 Fig. Identifiability and sensitivity analysis for the CD4 T cell lineage.** See the caption of S7 Fig for details. In this case we have $d = 11$ populations, and for panel D we simulated and fitted with model III.
(PDF)

**S9 Fig. Comparison of results from the integrated and sequential approaches.** Results are based on data from $n = 27$ mice. The sequential (SA) and integrated (IA) approaches both assign each cell to a subpopulation, but for any given cell this assignment may differ between the approaches. To quantify the similarity, we calculated the Jaccard index for each pair of subpopulations (one from the IA and the other from the SA). The Jaccard index is ratio of the number of cells that are in both clusters, and the number of cells that are in either one of the clusters. **A.** Results for CD8 T cell data. **B.** Results for CD4 T cell data.
(PDF)

**S10 Fig. Cluster similarity informed prior distribution on the $Q$-matrix. A.** Marginal posterior density of the mean differentiation rate $\mu_Q$. **B.** Marginal posterior density of the weight $\alpha_Q$ of the distance matrix $D_{ij} = \|\mu_i - \mu_j\|_2$ on the differentiation matrix elements $Q_{ij}$. **C.** and **D.** 95% credible intervals (lines) and posterior medians (dots) of $Q_{ij}$ as a function of the distance between the mixture components $i$ and $j$ in the latent space.
(PDF)

**S11 Fig. Pre-processing CD4$^+$ T cell flow cytometry data for the sequential approach.** Results are based on data from $n = 27$ mice. **A.** Marker expression heatmap for selected markers and consensus T-cell populations. **B.** UMAP of the marker expression data, colored by annotation. **C.** UMAPs of marker expression data, split by day post infection (DPI). The color scale reflects cell density in UMAP space. **D.** Time series of the fraction of cells in each cluster. The lines show a linear interpolation on the log scale.
(PDF)

**S12 Fig. Model fits to CD4 T cell time series using the sequential approach.** Results are based on data from $n = 27$ mice. **A.** Data and predictions from fitted models. Top panels show total antigen-experienced CD4 T cell counts in the lung, other panels show relative population sizes of each of the subpopulations. **B.** Parameter estimates using model II. **C.** Parameter estimates using model III. **D.** Estimated differentiation matrix in model III.
(PDF)

**S13 Fig. Posterior predictive checks for the integrated approach with CD4 T cell data.** Results are based on data from $n = 27$ mice. **A.** Marginal distributions of marker expression (cf. Fig 1, panel I). Data is shown in black, simulated data is shown in red. **B.** Input data and reconstruction using the autoencoder model. The number in brackets is the coefficient of determination ($R^2$).
(PDF)

**S14 Fig. Model fits to cell count data alone.** Results are based on data from $n = 27$ mice. **A.** time-homogeneous model with a single compartment (i.e. a log-linear model) fit to CD8 T-cell count data (dots). The model fit is shown as a blue curve (posterior median), with 95% CrI as a dark-blue band. The light-blue band shows the posterior predictive interval (i.e. simulated observations). **B.** Fit of model with a single compartment, but with time-dependent net loss rates $\lambda(t)$. **C.** Fit of a time-homogeneous model with two compartments. The population sizes (posterior median) of the two populations are shown as black curves. **D–F.** Fits of the three models to CD4 T cell counts.
(PDF)

**S15 Fig. The effect of batch correction in the integrated approach.** Results are based on data from $n = 27$ mice. Shown is the entropy of the experimental batch distribution around each

cell, using the latent vector $z$ and its nearest neighbors. Values for batch-corrected latent vectors are shown in blue. Mock corrected values are shown in orange, and values for randomized batch information are shown in green. Panels A and B show CD8 and CD4 data, respectively. The distributions are capped at 1.5 as there was a very small number of cells with high relative entropy.
(PDF)

## Author contributions

**Conceptualization:** Christiaan H. van Dorp, Donna L. Farber, Andrew J. Yates.

**Data curation:** Joshua I. Gray.

**Formal analysis:** Christiaan H. van Dorp, Joshua I. Gray, Daniel H. Paik.

**Funding acquisition:** Donna L. Farber, Andrew J. Yates.

**Investigation:** Christiaan H. van Dorp, Joshua I. Gray, Daniel H. Paik, Donna L. Farber, Andrew J. Yates.

**Methodology:** Christiaan H. van Dorp, Joshua I. Gray, Daniel H. Paik, Donna L. Farber, Andrew J. Yates.

**Project administration:** Donna L. Farber, Andrew J. Yates.

**Resources:** Andrew J. Yates.

**Software:** Christiaan H. van Dorp.

**Supervision:** Donna L. Farber, Andrew J. Yates.

**Validation:** Christiaan H. van Dorp, Joshua I. Gray.

**Visualization:** Christiaan H. van Dorp.

**Writing – original draft:** Christiaan H. van Dorp, Joshua I. Gray, Andrew J. Yates.

**Writing – review & editing:** Christiaan H. van Dorp, Donna L. Farber, Andrew J. Yates.

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
