## [Decision Letter · Decision Letter 0]

21 Apr 2025

PCOMPBIOL-D-25-00373

A variational deep-learning approach to modeling memory T cell dynamics

PLOS Computational Biology

Dear Dr. Yates,

Thank you for submitting your manuscript to PLOS Computational Biology. After careful consideration, we feel that it has merit but does not fully meet PLOS Computational Biology's publication criteria as it currently stands. Therefore, we invite you to submit a revised version of the manuscript that addresses the points raised during the review process.

Please submit your revised manuscript within 60 days Jun 21 2025 11:59PM. If you will need more time than this to complete your revisions, please reply to this message or contact the journal office at ploscompbiol@plos.org. Please include the following items when submitting your revised manuscript:

We look forward to receiving your revised manuscript.

Kind regards,

Rustom Antia

Academic Editor

PLOS Computational Biology

Mark Tanaka

Section Editor

PLOS Computational Biology

**Additional Editor Comments (if provided):**

**Journal Requirements:**

**Reviewers' comments:**

Reviewer's Responses to Questions

**Comments to the Authors:**

Reviewer #1: In this manuscript, the authors have addressed the challenges of confronting established mathematical modeling techniques with high-dimensional immunological data. The authors have used two different approaches, a sequential and an integrated deep learning-based approach, to understand the heterogeneity during the development and persistence of memory T cells after IAV infection in mice. They have discussed the identifiability issues of model structure and parameters using the sequential approach and used an advanced variational autoencoder model to tackle those issues in the integrated approach. Using the integrated approach, they have found that TRM heterogeneity is maintained long-term by a Bcl-2hi population of cells that can differentiate into functionally distinct subsets. Overall, the ideas presented in the manuscript were interesting and the problems identified are important. Below are some major and minor comments that should be addressed before publication,

Major comments:

Since the major conclusions of the paper is primarily based on the integrated approach, I think that the sequential approach part can be condensed down to only 1 figure, rather than 3 figures, with rest of the figures being in the SI. The main figure can just show the clustering part of the flow cytometry data to identify the subsets and 4 model fits to the trajectory data of different subsets.

The authors have done a good job of describing the issues with the sequential approach. However, I found it difficult to interpret how they have solved these problems of the sequential approach using the integrated approach. For example, in lines 360-370, they talk about the identifiability issues of the sequential approach and how they addressed this in the integrated approach. But, what’s the ultimate conclusion of this part?

Further, the idea and rational behind the Gaussian mixer model used in the integrated approach needs to be explained in detail. Why use this model?

Also, the authors have mentioned in the manuscript that the generated and predicted cluster frequencies are interdependent in the integrated approach and that can result in better fits. This is somewhat concerning to me even though the authors have tried to validate this approach.

Minor comments:

In Fig 1D-G, what are the points and the lines? for each time point, why are there 3-4 points? Do the lines represent a fitted curve already or just mean trends?

Please explain the Leiden clustering method in more detail somewhere in the text or SI. How is this method different from traditional clustering methods? Also, does it result in different clustering patterns for different initial conditions for each run?

In Fig 3C, Instead of plotting the estimated decay rates, it might be helpful to visualize the estimated half lives from your estimated decay rates?

Please explain in detail how did you calculate the pattern in fig 5C from fig 5B?

In Fig 5E and 7G, it might be helpful to rename TRM1,2,3 with actual marker labels. Also what are the dots and the lines represent in these figures? Are the dots data and the lines fitted?

Reviewer #2: In their paper "A variational deep-learning approach to modeling memory T cell dynamics", van Dorp et al. provide an innovative approach to classify and track phenotypic clusters of resident memory T cells following an influenza infection. Their approach effectively analyzes high-dimensional data generated from low-throughput experiments to gain tractable insights into immunological responses. Through this approach, they offer a critical opportunity to identify and characterize processes underlying the regulation of various immunological responses. Indeed, this method appears applicable to a wide range of biological systems, significantly contributing to the insights gained from mathematical models and measured data. Additionally, this work provides insight into how advances in generative AI can benefit mathematical modeling in life sciences, and may lead to further adoption and development of computational methods.

As I have little knowledge in immunology, I cannot realistically assess the identifiability of cell differentiation reported in their model. However, it is important to point out that identifiability, especially with the insertion of prior knowledge in the form of modeling constraints, is a major concern in such analyses. The comprehensive analysis and discussion of these issues are, in my view, sufficient and of great use for the careful and considerate application of similar methods that will originate from this approach. Therefore, I only have minor comments:

Line 361: typo: ...learned by by the model...

Line 430-432: "We chose to continue with model III as it performed just as well as the full model and enabled us to further explore differentiation pathways" While I agree with this choice, specifically because it performs _just as well_, I don't think the reasoning here is sufficiently sound. This also relates to Line 249-251, as there you identify that Model IV can unsurprisingly overfit Model III.

**Have the authors made all data and (if applicable) computational code underlying the findings in their manuscript fully available?**

Reviewer #1: None

Reviewer #2: Yes

PLOS authors have the option to publish the peer review history of their article (what does this mean?). If published, this will include your full peer review and any attached files.

Reviewer #1: No

Reviewer #2: **Yes: **Nicolas Ochsner

**Figure resubmission:**
---

## [Decision Letter · Decision Letter 1]

15 Jun 2025

Good work Andy and team for a hard problem.

Rustom

Dear Prof. Yates,

We are pleased to inform you that your manuscript 'A variational deep-learning approach to modeling memory T cell dynamics' has been provisionally accepted for publication in PLOS Computational Biology.

Best regards,

Rustom Antia

Academic Editor

PLOS Computational Biology

Mark Tanaka

Section Editor

PLOS Computational Biology

Reviewer's Responses to Questions

**Comments to the Authors:**

Reviewer #1: The authors have addressed the comments that I raised earlier.

**Have the authors made all data and (if applicable) computational code underlying the findings in their manuscript fully available?**

Reviewer #1: Yes

PLOS authors have the option to publish the peer review history of their article (what does this mean?). If published, this will include your full peer review and any attached files.

Reviewer #1: **Yes: **Ananya Saha

---

## [Editor Report · Acceptance letter]

PCOMPBIOL-D-25-00373R1

A variational deep-learning approach to modeling memory T cell dynamics

Dear Dr Yates,

I am pleased to inform you that your manuscript has been formally accepted for publication in PLOS Computational Biology. Your manuscript is now with our production department and you will be notified of the publication date in due course.

With kind regards,

Zsuzsanna Gémesi
